
**Future water storage changes over the Mediterranean, Middle East, and North Africa in response to global warming and stratospheric aerosol intervention**

**Abolfazl Rezaei[1,2], Khalil Karami[3], Simone Tilmes[4], John C. Moore[5]**

[1] Department of Earth Sciences, Institute for Advanced Studies in Basic Sciences (IASBS), Zanjan
45137-66731, Iran. arezaei@iasbs.ac.ir; abolfazlrezaei64@gmail.com.
[2] Center for Research in Climate Change and Global Warming (CRCC), Institute for Advanced Studies
in Basic Sciences (IASBS), Zanjan 45137-66731, Iran
[3] Institut für Meteorologie, Stephanstraße 3, 04103 Leipzig, Germany. khalil.karami@uni-leipzig.de
[4] National Center for Atmospheric Research, Boulder, CO, USA. tilmes@ucar.edu
[5] Arctic Centre, University of Lapland, Rovaniemi, 96101, Finland. john.moore.bnu@gmail.com
Corresponding Author: Abolfazl Rezaei, arezaei@iasbs.ac.ir; abolfazlrezaei64@gmail.com.
**Abstract**
Water storage plays a profound role in the lives of people across the Middle East and North Africa
(MENA) as it is the most water stressed region worldwide. The lands around the Caspian and
Mediterranean Seas are simulated to be very sensitive to future climate warming. Available water
capacity depends on hydroclimate variables such as temperature and precipitation that will depend
on socioeconomic pathways and changes in climate. This work explores changes in both the mean
and extreme terrestrial water storage (TWS) under an unmitigated greenhouse gas (GHG) scenario
(SSP5-8.5) and stratospheric aerosol intervention (SAI) designed to offset GHG-induced warming
above 1.5 °C and compares both with historical period simulations. Both mean and extreme TWS are
projected to significantly decrease under SSP5-8.5 over the domain, except for the Arabian Peninsula,
particularly in the wetter lands around the Caspian and Mediterranean Seas. Relative to global
warming, SAI partially ameliorates the decreased mean TWS in the wet regions while it has no
significant effect on the increased TWS in drier lands. In the entire domain studied, the mean TWS is
larger under SAI than pure GHG forcing, mainly due to the significant cooling, and in turn, a
substantial decrease of evapotranspiration under SAI relative to SSP5-8.5. Changes in extreme water
storage excursions under global warming are reduced by SAI. Extreme TWS under both future
climate scenarios are larger than throughout the historical period across Iran, Iraq, and the Arabian
Peninsula, but the response of the more continental eastern North Africa hyper-arid climate is
different from the neighboring dry lands. In the latter case, we note a reduction in the mean TWS

trend under both GHG and SAI scenarios, with extreme TWS values also showing a decline compared to historical conditions.

**Keywords:** Mean and extreme water storage; SSP5-8.5; Stratospheric Aerosol Intervention; Global warming; MENA region, Caspian and Mediterranean Seas

**500-character short summary**

Water storage (WS) plays a profound role in the lives of people in the Middle East and North Africa and Mediterranean climate "hot spots". Simulated is WS changed by greenhouse gas (GHG) warming with and without stratospheric aerosol intervention (SAI). WS significantly increases in the Arabian Peninsula and decreases around Mediterranean under GHG. While SAI partially ameliorates the GHG impacts, projected WS increases in dry regions and decreases in wet areas relative to the present climate.

## 1. Introduction

The Middle East and North Africa (MENA), with 6% of the world's population, are currently among the most water-stressed regions worldwide (Fragaszy et al., 2020). The dry climate, intensifying droughts, increasing population, and water over-extraction particularly across the Middle East (World Bank, 2017), make it home to 12 of the 17 most water-stressed countries on the planet (Hofste et al., 2019). Water availability is crucial for sanitation (Reiter et al., 2004), economic activity (UNESCO, 2003), ecosystems (Shiklomanov and Rodda, 2003), and hydrological systems (Mooney et al., 2005).

The MENA region has the largest expected economic losses from climate-related water scarcity, robustly estimated at 6–14 % of Gross Domestic Product (GDP) by 2050 (World Bank, 2017). MENA's terrestrial water storage (TWS) is being intensively extracted and may act as a flashpoint for conflict (Famiglietti, 2014). TWS incorporates all water on the land surface (snow, ice, water stored in the vegetation, river, and lake water) and in the subsurface (soil moisture and groundwater). Beyond anthropogenic activities, natural climate variability such as drought frequency affects water storage and agriculture, which then impacts food security (Fragaszy et al., 2020). The Middle East is especially prone to severe and sustained droughts due to its location in the descending limb of the Hadley circulation and associated dry and semiarid climate (Barlow et al., 2016). The 1998-2012 14-year period was the worst drought in the past 900 years (Cook et al., 2016). Because the saturated vapor pressure of air is largely controlled by temperature, any change in temperature, as well as

precipitation, substantially affects (Konapala et al., 2020; Ajjur and Al-Ghamdi, 2021; Hobeichi et al., 2022) the water storage capacity available to supply the increasing water demand in the region (Lian, 2021). The MENA region, having both low precipitation and high evaporation, is very vulnerable to climate change (Giorgi, 2006; Lelieveld et al., 2012; Tabari and Willems, 2018; Zittis et al., 2019). MENA water storage is therefore particularly sensitive to any perturbation of the water cycle imposed by global warming.

GHG warming has already adversely affected water resources in the MENA region (Wang et al., 2018) and is simulated to intensify water competition between states (Arnell, 1999) in the future. Although global warming is expected to increase precipitation and soil moisture across MENA (Cook et al., 2020), it will decrease runoff and groundwater recharge by larger amounts (Milly et al., 2005; Shaban, 2008; Suppan et al., 2008). Using the GHG emission scenario A1B simulated by nine CMIP3-class climate models, Droogers et al. (2012) projected that 22% of the future annual water shortage, 199 $km^3$ in 2050 in MENA, will be due to global warming. 17 global climate models from Coupled Model Intercomparison Project Phase 6 (CMIP6) under SSP5-8.5 simulate a significant increase in precipitation (+0.05 to $0.3 \mp 0.1$ mm day$^{-1}$) over South-Eastern Saharan Desert in NA by the end of the century (Arjdal et al., 2023). They also projected that the total soil moisture would increase over Southern Saharan Desert under the SSP5-8.5 (6 to 20%) and SSP2-4.5 (4 to 14%). Based on TWS data from eight global climate models participating in CMIP6, a broad part of the dry MENA region tends to be wetter under SSP5-8.5 over 2071-2100 (Xiong et al., 2022). GHG-driven groundwater storage depletion in the Middle East during the 21st century will far exceed that during the 20th century due to the increased evapotranspiration (ET) and reduced volume of snowmelt (Wu et al., 2020).

Although MENA's adjacent densely populated region, the Mediterranean, has a better water storage state, it is projected to substantially suffer from reduced water availability under future GHG climate scenarios (Lionello et al., 2006). This is due to both projected significant decreases in rainfall (MedECC, 2020) and large increases in demand for irrigation water by the end of the 21st century (Fader et al., 2016). The precipitation and water availability in the Mediterranean region, to the northwest of the MENA, is also projected to be highly sensitive to global warming, particularly regarding water availability (Lionello et al., 2006), having the largest differences in the water availability between 1.5 and 2°C warming scenarios globally (Schleussner et al., 2016). Global warming decreases Mediterranean groundwater recharge according to simulations under the IPCC A2 and B2 scenarios simulated using ECHAM4 and HadCM3 models (Döll and Flörke, 2005). Runoff

is decreased by 10-30% according to 12 models such as CCSM3, and ECHAM5/MPI-OM (Milly et al.,
2005), and soil moisture z-scores (obtained by taking the difference from the average and then
dividing it by the standard deviation of the time series from the baseline period) by -1 to -4 in warm
seasons according to simulations under SSP1-2.6, SSP2-4.5, SSP3-7.0, and SSP5-8.5 (Cook et al.,
2020). Water availability in turn is lowered by 8-28% for a warming of 2 °C as simulated by 11
CMIP5-class models by Schleussner et al. (2016). Likewise, Döll et al. (2018) found a strong drying in
the Mediterranean region under global warming since the largest precipitation decreases worldwide
were simulated in this region under SSP1-2.6, SSP2-4.5, SSP3-7.0, and SSP5-8.5 scenarios (Cook et
al., 2020). CMIP5 model results also confirm that the global warming (RCP2.6 and RCP6.0)
substantially decreases the TWS in the Mediterranean by the mid- (2030-2059) and late- (2070-
2099) twenty-first century (Pokhrel et al., 2021).

If global mean surface temperature rises to exceed 1.5 °C above the preindustrial mean temperature,
severe global consequences, and societal problems can be expected (Masson-Delmotte, 2022). Solar
radiation modification (SRM), a form of intervention to cool the climate by reflecting sunlight, has
been proposed as a potential method of limiting global temperature rises and the associated impacts
of increased GHG emissions. SRM may be the only way to keep or reduce surface temperatures to 1.5
°C given the reality of the GHG mitigation measures that have been agreed upon to date (MacMartin
et al., 2022). Simulations have shown a 2% decrease in total solar irradiance roughly offsets global
warming due to a doubling of $CO_2$ concentrations, and continuous injections of 10-18 Tg $SO_2$ per year
would lead to a cooling of about 1 °C after several years (WMO, 2022).  This is consistent with
observed surface cooling after large volcanic eruptions, such as the 1991 Mt Pinatubo eruption which
produced cooling of about 0.3 °C over a 2-3 year period (e.g., IPCC, 2021).

Many global climate models have simulated SRM in the form of stratospheric aerosol intervention
(SAI). Model studies include the Stratospheric Aerosol Geoengineering Large Ensemble Project
GLENS (e.g., Cheng et al., 2019; Simpson et al., 2019; Abiodun et al., 2021), the Geoengineering Model
Intercomparison Project (Kravitz et al., 2013; Tilmes et al., 2013), as well as others (e.g., Bala et al.,
2008; Jones et al., 2018; Muthyala et al., 2018). Compared with global warming, SAI decreases mean
global precipitation (Govindasamy and Caldeira, 2000; Bala et al., 2008; Robock et al., 2008; Cheng
et al., 2019; Simpson et al., 2019) as well as both the intensity and frequency of precipitation extremes
caused by GHG-induced climate change (Tilmes et al., 2013; Muthyala et al., 2018). Dagon and Schrag
(2016) is a rare article that focuses on the spatial variability of runoff and soil moisture responses to
SRM. Although solar geoengineering weakens the global hydrologic cycle (e.g., Bala et al., 2008;
Tilmes et al., 2013; Ricke et al., 2023), its regional impacts are method- and strategy-dependent
(Ricke et al., 2023) with potentially substantial changes in the regional precipitation patterns (Ricke
et al., 2010; Tilmes et al., 2013; Crook et al., 2015; Dagon and Schrag, 2016, Tilmes et al., 2020). While
differences in temperature fields vary relatively smoothly with radiative forcing, precipitation
patterns are far more variable being dependent on atmosphere/ocean/land surface coupling on a
wide range of spatial and temporal scales. Furthermore, SAI simulations rely on many model-specific
details and parameterizations that tend to produce larger across-model differences than simulations
using simpler forms of SRM (Visioni et al., 2021).  While SAI may counteract the annual-mean water
availability changes over land forced by GHG, it is not easy to offset the regional consequences,
especially in the hydrological cycle, such as the Amazonian drying trend and its reduced precipitation,
evaporation, and precipitation minus evaporation (Jones et al., 2018).

Although the MENA region and the adjacent Mediterranean region are known to be a "hot spot" for
climatic change (Giorgi and Lionello, 2008; Bucchignani et al., 2018), little has been done on potential
changes in TWS across MENA especially under SRM climates. This study fills that knowledge gap and
explores the changes that may occur in TWS under i) a high GHG emissions scenario, ii) the same GHG
scenario combined with SAI designed to globally neutralize the GHG radiative forcing, and iii)
compares both future climates with the historical conditions (1985-2014) across the Mediterranean,
Middle East, and northern Africa (NA).

**2. Data and Methods**
**2.1. Study Area**

The study area is composed of MENA and southern Europe to its north including the Caspian

and Mediterranean Seas. MENA covers the large region from Morocco in the west to Iran in the east,
containing all the Maghreb and the Middle Eastern countries from the $15^0$N to $45^0$N latitude and from
$20^0$W to $63^0$E longitude (Fig. 1). As well as a water-stressed region, MENA, is a worldwide hot spot
for exacerbated extreme temperatures, aridity conditions, and drought (Giorgi and Lionello, 2008;
Bucchignani et al., 2018). According to the Koppen Climate Classification System (Peel et al., 2007),
MENA broadly has a hot and arid climate except for the coastal regions and highlands. Most of NA has
a desert climate and 90% is covered by the Saharan Desert. The 2 m air temperature rises to 50°C in
summertime while the annual mean precipitation is less than 25 mm (Faour et al., 2016). The Arid
Steppe climate predominates in Morocco, Algeria and Tunisia with cold winters (Faour et al., 2016)
except for the Atlas Mountains which are cooler and wetter (annual mean precipitation of ~500 mm).

Across the Middle East, the largest amount of precipitation falls in four main regions: the

coastal eastern Mediterranean Sea, the south coast of the Caspian Sea, the western sides of the Zagros
Mountains across Iran and Iraq, and the southern tip of the Arabian Peninsula. The Middle East also
contains several major deserts having little to no precipitation: the Lut and Kavir deserts in the south-
east and north-central regions in Iran, the Arabian Desert, the Syrian Desert, and the Negev in south-
eastern corner of the Mediterranean Sea. Middle East precipitation often originates from moisture
coming from the west over the Mediterranean Sea (Evans and Smith, 2006). The Red Sea and the
Persian Gulf are also source regions for the heaviest precipitations across the area.

The Mediterranean area has mild wet winters and warm to hot, dry summers as well as a

complicated morphology, owing to the many steep orogenic structures, distinct basins and gulfs,
along with islands and peninsulas of various sizes (Lionello et al., 2006).

Based on its full range of climate types, we divided the study area into six sub-regions (R1 to

R6) to explore the changes in hydroclimate variables under both global warming and SAI scenarios
(Fig. 1). The regions R1 to R6 respectively refer to the lands around the Caspian Sea, eastern Middle
East (largely containing Iran and Iraq), Mediterranean area, Arabian Peninsula, eastern NA, and
western NA. The simulated present-day climatology (1985-2014) of each region for different
hydrological quantities is summarized in Table 1. Potential evapotranspiration (ET) is the amount of
evaporation that would occur if a sufficient water source were available. The Thornthwaite method
was used to calculate the potential ET based on the monthly mean temperature and latitude data for
each grid. Evaporation from both soil and canopy and transpiration are summed up to obtain the real
ET, which is the quantity of water actually removed from a surface by evaporation and transpiration.
The lands around the Caspian and Mediterranean Seas with a cooler climate, have the highest
precipitation and real ET while more continental eastern NA with hyper-arid climate (with annual
precipitation less than 100 mm) has the lowest precipitation, real ET, soil moisture, and TWS. The
lands around the Caspian Sea have the highest soil moisture and TWS. More continental refers to an
area with characteristics that are typical of continental climates and is less influenced by the
moderating effects of nearby oceans.

**Table 1.** The medians of precipitation, temperature, real evapotranspiration (ET), soil moisture, terrestrial water storage (TWS), and potential ET over each region (R1 to R6, see Fig. 1) during the historical period according to the model outputs. The results for global warming and SAI are further shown in Table S1.

| Region | R1 | R2 | R3 | R4 | R5 | R6 |
|---|---|---|---|---|---|---|
| Precipitation (mm/yr) | 321 | 182 | 479 | 78 | 48 | 112 |
| Temperature ($^0$C) | 14.2 | 20.5 | 17.2 | 27.0 | 23.7 | 25.3 |
| Real ET (mm/yr) | 419 | 187 | 388 | 72 | 50 | 112 |
| Soil moisture (Kg/m$^2$) | 1846 | 1771 | 1572 | 1353 | 1155 | 1287 |
| TWS (Kg/m$^2$) | 2091 | 1776 | 1623 | 1348 | 1167 | 1313 |
| Potential ET (mm/yr) | 74 | 123 | 74 | 210 | 143 | 185 |

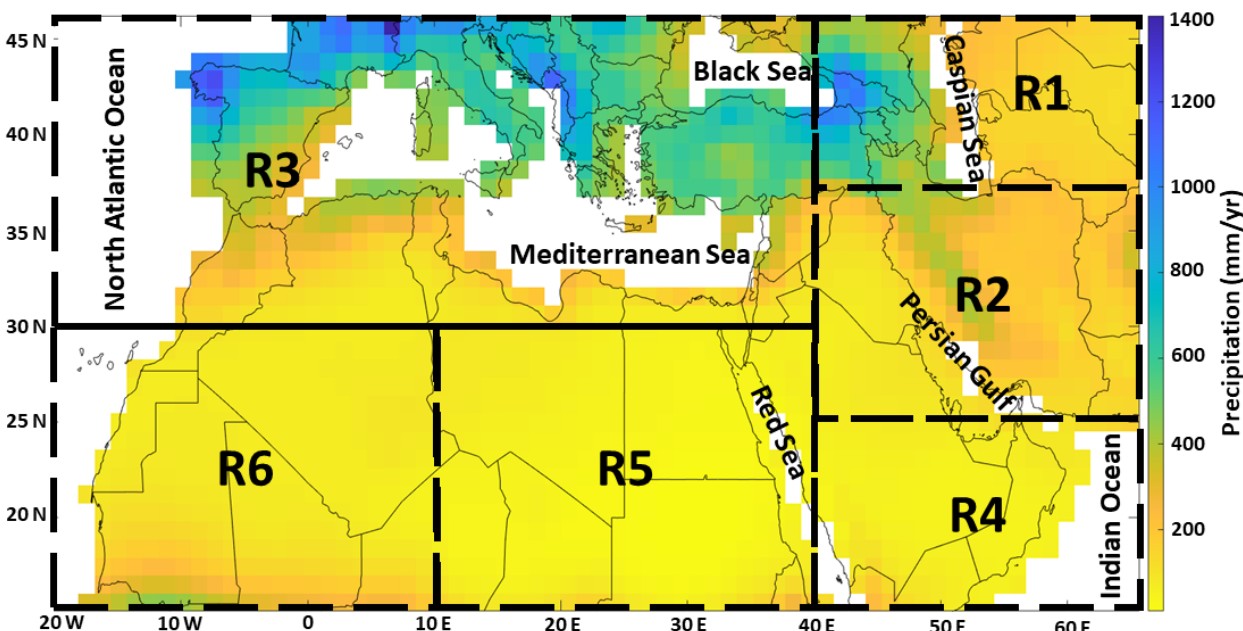

**Figure 1.** The MENA's annual precipitation map during the historical period. Regions R1 to R6 largely refer to the lands around the Caspian Sea, the eastern Middle East (largely containing Iran and Iraq), the Mediterranean area, Arabian Peninsula, eastern North Africa (NA), and western NA, respectively.

## 2.2. Model simulations and scenarios

We examined the data from the NCAR Community Earth System Model version 2- Whole Atmosphere Community Climate Model Version 6 (CESM2(WACCM6)) that simulated the CMIP6 (Eyring et al., 2016) scenarios. CESM2 ranks among the top nine models known for their accuracy in simulating

global precipitation patterns, based on the Hellinger distance metric, which compares the bivariate
empirical densities of CESM2 with those of 34 CMIP6 models, against historical precipitation data
sourced from the Global Precipitation Climatology Centre (GPCC) (Abdelmoaty et al., 2021). CESM2
has precipitation biases about 20% lower than CESM1 (Danabasoglu et al., 2020). CESM2(WACCM6)
has an interactive stratospheric aerosol treatment (Danabasoglu et al., 2020) that is consistent with
observations (Mills et al., 2016). For global terrestrial ET, the CESM2(WACCM6) ranked as the second-
best model among 19 CMIP6 models (Wang et al., 2021). Furthermore, CESM2(WACCM6), reproduced
the observed global land carbon trends remarkably well (Danabasoglu et al., 2020), and includes a
full ocean model (Parallel Ocean Program version 2, POP2) to simulate the response of stratospheric
aerosol change in the climate.

CESM2 also demonstrates satisfactory performance in simulating historical climate conditions within the
study area. In the evaluation by Babaousmail et al. (2021), which assessed 15 CMIP6 models in replicating
monthly rainfall patterns spanning from 1951 to 2014 in NA, CESM2(WACCM6) emerged as one of the
top-performing models. It accurately captured rainfall peaks across the region, albeit with a slight
overestimation (ranging from 5 to 10 mm/month) in the southern areas and a slight underestimation
(ranging from 0 to 20 mm/month) in the northern regions. Despite these minor deviations,
CESM2(WACCM6) was recognized as one of the models for well simulating precipitation patterns across
NA, achieving a Taylor skill score of 0.62. Evaluation of CESM2(WACCM6) across the Mediterranean coasts
placed it at the $9^{th}$ and $17^{th}$ positions out of 31 CMIP6 models for its performance in simulating
temperature and precipitation (Bağçaci et al., 2021). Furthermore, when it comes to simulating
precipitation relative to observational data for northeastern Iran during the period of 1987-2005, CESM2
stood out as the top-performing model among six CMIP6 models (Zamani et al., 2020). Assessing the
representation of spatial and temporal variations in historical precipitation from 1980 to 2014 across
Africa and the Arabian Peninsula, the CMIP6 multi-mean ensemble (inclusive of CESM2(WACCM6))
demonstrated reasonable performance, as highlighted in Nooni et al. (2023).

The SAI simulation we use (SSP5-8.5-SAI) is designed to employ SAI together with the high GHG
emissions scenario, SSP5-8.5 with the target of limiting the mean global temperatures to 1.5°C above
the pre-industrial (1850–1900) conditions (Tilmes et al., 2020). Under SSP5-8.5 forcing, Tilmes et al.
(2020) projected this threshold is exceeded around the year 2020 in CESM2(WACCM6). The
atmospheric component of CESM2(WACCM6) has a resolution of 1.25° in longitude and 0.9° in
latitude. The experiment injects $SO_2$ at 180° longitude at four predefined latitudes (30°N, 30°S, 15°N,
and 15°S) at around 25 km in 15°N/S and around 22 km at 30°N/S as suggested by Tilmes et al.
(2018), using a feedback control algorithm to maintain not just the global mean temperature, but the
interhemispheric and equator-to-pole temperature gradients (Tilmes et al., 2020). For SSP5-8.5-SAI,
most of the sulfur mass was injected at 15°S, some at 15°N and 30°S, and very little at 30°N. We used
the monthly TWS (the sum of snow water equivalent and soil moisture (Wu et al., 2021)),
precipitation, temperature, water evaporation from soil and canopy, transpiration, soil moisture, and
leaf area index (LAI) data from all five ensemble members (r1 to r5) of the SSP5-8.5 scenario and the
three available ensemble members (1-3) of SSP5-8.5-SAI. The results for variables other than TWS
are shown in the Supplementary Information. For the historical period, we used all three available
realizations (r1 to r3) from CESM2(WACCM6). For the anomaly analysis relative to historical
conditions and the multiple linear regression models, we used the first three ensembles of SSP5-8.5,
consistent with the three available historical members. We compare the GHG and SAI scenarios over
2071-2100 with the 1985-2014 historical period.

We focused on the historical period from 1985 to 2014 rather than the entire historical dataset spanning
from 1850 to 2100 for several reasons. Firstly, recent historical climate data may exhibit less uncertainty,
given that additional meteorological stations with improved data quality are available to be used for
model calibrations (Zhang et al., 2020). Secondly, this selected historical period offers valuable insights
into the observable impacts of climate change, which are highly pertinent to present-day societal and
environmental challenges. These insights are of utmost importance to policymakers and communities
alike. Thirdly, the chosen historical 30-year time period aligns with the 30-year periods considered for the
GHG emissions and SAI scenarios, ensuring consistency in our statistical analysis. We focus on the 2071-
2100 future period because the anticipated changes in TWS driven by GHG emissions are expected to be
more pronounced during this time frame (Pokhrel et al., 2021). Furthermore, the SAI forcing is strongest
in the later period of the simulation and is expected to produce a more significant result.

**2.3. Return periods**
We are interested in climate extremes, not only changes in means. Therefore, we examine how the
frequency of events of some particular levels are likely to change under different scenarios. We use
the generalized extreme value (GEV) distribution function to estimate the probability distribution
function of the TWS extremes. A return period is an estimated average time between events such as
floods or river discharge flows. It is calculated by generating the 95% normal-approximate
confidence intervals in accordance with the mean and variance of the variable (here TWS).
The GEV probability density and cumulative distribution functions are defined as (Gilleland, 2020):
$$g(z) = \frac{1}{\sigma} t(z)^{1+\xi} e^{-t(z)}; \quad G(z) = e^{-t(z)}; \quad t(z) = \begin{cases} \left\{ 1+\xi\left(\dfrac{z-\mu}{\sigma}\right) \right\}^{-1/\xi}, & \xi \neq 0 \\[2mm] e^{-\left(\frac{z-\mu}{\sigma}\right)}, & \xi = 0 \end{cases} \tag{1}$$
For $\xi \neq 0$, we have $t(z)^{1+\xi} = \left\{ 1+\xi\left(\dfrac{z-\mu}{\sigma}\right) \right\}^{-(1+1/\xi)}$ and for $\xi = 0$, the $z$ domain restricted to
$\xi\left(\dfrac{z-\mu}{\sigma}\right) > -1$. The GEV distribution is parameterized using $\xi$, $\mu$, and $\sigma$ which are the shape,
location, and scale parameters, respectively and analogous to the skewness, mean and standard
deviation. We assume that the GEV is the valid distribution function for variables $z_1, \ldots, z_n$
representing the annual maximum return TWS levels, where the quantiles of the distribution
function give the return levels, $z_p$. The return levels are the solutions to $G(z_p) = 1-p$, which yields
(Gilleland, 2020):
$$z_p = \begin{cases} \mu - \dfrac{\sigma}{\xi}[1-\{-\ln(1-p)\}^{-\xi}] & \text{for } \xi \neq 0 \\[2mm] \mu - \sigma \ln\{-\ln(1-p)\} & \text{for } \xi = 0 \end{cases} \tag{2}$$
where $p$ is probability corresponding to $z_p$. The return period is obtained as:
$\quad return\ period\ (i) = 1/(1-cdf\ (i)) \tag{3}$
where $cdf$ is the cumulative distribution function. We also calculated the 95% asymptotic lower and
upper confidence intervals based on the Kolmogorov-Smirnov statistic (Doksum and Sievers, 1976).
We used the concatenated TWS anomaly data for the historical period, high GHG emissions, and SAI
scenarios to analyze the return periods. As an example, the relationship between empirical quantiles
and model quantiles as well as the probability density versus quantiles for the regions R2 and R5 are
shown in Figs. S1 and S2.

## 2.4. Multiple linear regression (MLR) model

We want to analyze how the primary driving climate fields (surface air temperature, precipitation,
ET, and LAI (i.e., vegetation coverage)) for TWS vary spatially and among the different scenarios
(Zhang et al., 2022). We use a simple multiple linear regression (MLR) model with TWS as the
dependent variable (Y) for each ensemble member in each region. The following procedures were
conducted:
i) Employing the variable clustering (VARCLUS) procedure to thoroughly assess collinearity among
the variables. VARCLUS is a method that effectively segregates a set of numeric variables into disjoint
or hierarchical clusters, each characterized by a linear combination of the variables within the cluster
(Sarle, 1990). The criterion is that when the proportion of the variance explained by a cluster is larger
than 0.8, it is advisable to select one variable from that cluster. Based on the results obtained from
VARCLUS (Figs. S3 and S4), we made specific decisions to enhance the robustness of our analysis. For
instance, we identified strong correlations exceeding 0.9 between potential ET and temperature
(Tables S2-S13 in the Supplementary Information), as well as between soil moisture and TWS in all
cases (except for the eastern NA (R5) in Tables S2-S13). Consequently, we chose to exclude potential
ET and soil moisture from our analysis due to their high levels of correlation with temperature and
TWS, respectively.
ii) Considering a linear regression model with potential independent variables (X): temperature,
precipitation, real ET, and LAI. We conducted a temporal autocorrelation analysis on all these
independent variables for each model. This analysis was carried out using the autocorrelation
function at a 95% confidence level. In all regions (except R4), the autocorrelation results indicated
that the lags at the first and second months were statistically significant, while the third month lag
was almost non-significant. Therefore, we modified the MLR model to include information from the
two preceding months in these regions. However, in region R4, we observed different patterns. In this
region, both real ET and temperature significantly depended on their respective conditions from the two
previous months, while precipitation did not show this effect. Moreover, LAI in R4 exhibited
dependencies on the first three and four preceding months under the SSP5-8.5 and SSP5-8.5-SAI
scenarios, respectively. Consequently, we incorporated specific lagged months for each variable in
R4.
iii) Identifying the outliers using the Bonferroni *p*-values (i.e., Bonferroni correlation) and then
removing them. Bonferroni correlation is a modification for *p*-values when several dependent or
independent statistical tests are being accomplished concurrently on a single data set.  A Bonferroni
correction divides the critical *p*-value by the number of comparisons being made (Bland and Altman,
1995). The number of outlier data points excluded varies from zero to 5 (over the 700 point) in the
36 models.
iv) Fitting the final model after removing the outliers. In all regions and scenarios, the MLR models
are statistically significant at the 95% level. The variance explained ($R^2$) varies from around 0.3 in
the dry southern MENA to 0.89 and 0.96 in the wetter lands around the Caspian and Mediterranean
Seas.
v) Assessing the relative "importance" of the variables for TWS in the final model using the Lindeman,
Merenda, and Gold (LMG) method (Lindeman et al., 1980), where the fractional variance accounted
for is determined as the independent variable-order average over average contributions in models
of different sizes. The LMG method considers the average contributions of each variable across
different model sizes and then averages these averages to provide a more robust measure of variable
importance. The LMG can be defined as (Grömping, 2007):
$$LMG(x_k) = \frac{1}{p!} \sum_{Permutation} seqR^2(\{x_k\} \mid r) \qquad (4)$$

where $seqR^2(\{x_k\} \mid S_k(r)) = R^2(\{x_k\} \cup S_k(r)) - R^2(S_k(r))$ and
$$R^2(S) = \frac{Model\ SS(\text{mod}\ el\ with\ regressors\ in\ set\ S)}{Total\ SS}$$

Orders have the same $S_k(r) = S$ summarize into a single summand, we therefore can re-write
Eq. (4):
$$LMG(x_k) = \frac{1}{p!} \sum_{S \subseteq \{x_1,\dots,x_p\} \setminus \{x_k\}} n(S)!(p - n(S) - 1)! seqR^2(\{x_k\} \mid S) \qquad (5)$$

LMG has been recommended by Johnson and LeBreton (2004) and Grömping (2007) since it uses
both direct effects and impacts adjusted for other regressors in the model. As the considered
variables may be correlated with each other, when a new predictor is added to a model that already
contains other predictors, its impact can be influenced by the presence of those other variables. The
LMG method takes into account these interactions and adjusts the variable's contribution to reflect
its unique impact while considering the effects of other regressors. Importance is a unitless variable
and the sum of all independent variable importance's in each model equals the model's explained
variance. Here we use all three ensemble members separately to estimate the robustness of the
importance estimates.

**3. Results:**
**3.1. Mean terrestrial water storage (TWS) changes due to GHG and SAI**
In this section, we present the projected changes in TWS across MENA and the lands around the
Caspian and Mediterranean Seas. We discuss trends in the TWS anomalies relative to TWS averaged
over the historical period (1985-2014) in response to both GHG (SSP5-8.5) forcing and to GHG+SAI.
Figure 2 illustrates the original TWS anomalies, while Fig. S5 exclusively presents the long-term

component, providing a clearer understanding of the changes under climate scenarios. The positive and negative anomalies in these figures refer to increasing and decreasing TWS, respectively. The trend decreases in the northern parts (R1 and R3) and eastern NA (R5) with a hyper-arid climate but rises in the Arabian Peninsula (R4) and western NA (R6) under both GHG and SAI scenarios, particularly over the latter part of the 21st century. In all regions, the SAI climate TWS is higher than SSP5-8.5 or at least lies in the across-range of SSP5-8.5 towards the end of the century, especially in R2 and R5 (Figs. 2 and S5). The TWS difference between SAI and global warming in the region R2, particularly over the latter part of the 21st century, is greater relative to the rest of the domain. The TWS change is smaller in the hyper-arid eastern NA (R5) than the other regions under both climate scenarios.

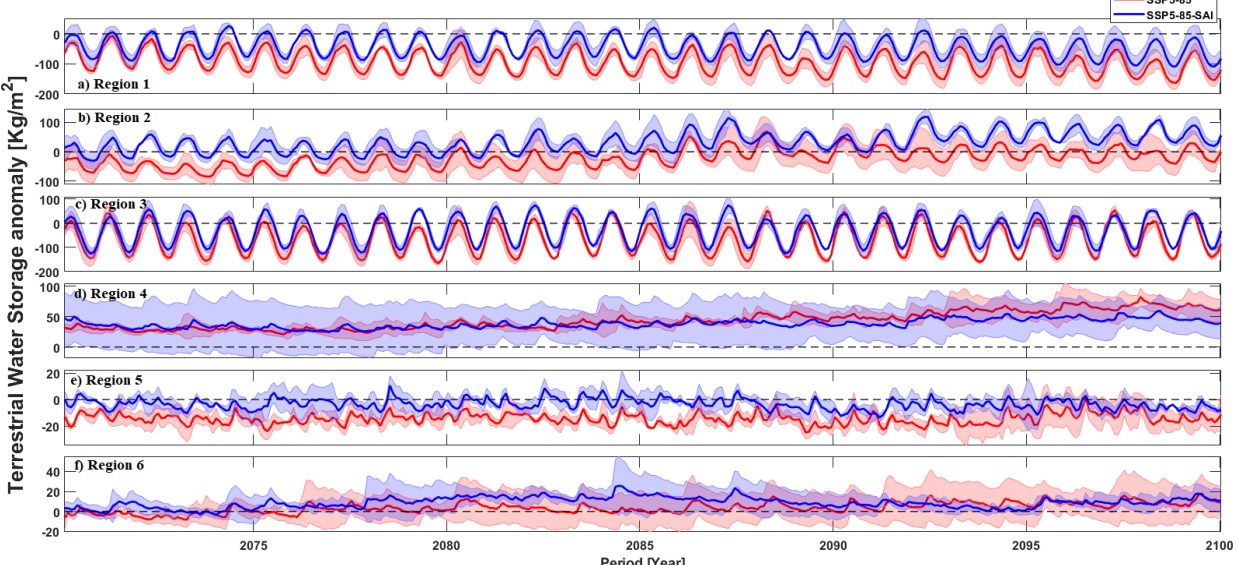

**Figure 2.** The TWS anomaly relative to the TWS averaged over the historical period across MENA and the lands around the Caspian and Mediterranean Seas under global warming without (SSP5-8.5) and with SAI (SSP5-8.5-SAI). Figures a-f respectively are for regions R1 to R6. Shading in each curve shows the across-ensemble range. The dashed line crossing the *y*-axis at zero in each subplot is the ensemble mean of TWS over the historical period (1985-2014).

Figure 3 depicts the TWS differences between the historical (1985-2014) and the future climate scenarios over the 2071-2100 period. Consistent with the above findings, Figs. 3b and S6a-c show that the TWS response to GHG forcing in the wet regions around the Caspian (R1) and Mediterranean (R3) Seas is simulated as declining, while across the (semi)arid MENA region, particularly in central Iran (R2), the Arabian Peninsula (R4), and the southern portions of NA (R5 and R6), there is a positive trend. Under global warming, the largest decrease in TWS occurs around the Caspian (particularly in the east) and the Mediterranean (except for its north) while its most robust increase happens in the

southern margins of NA and the eastern parts of the Arabian Peninsula. SAI (Figs. 3c and S6d, e, and f) partially counteracts the changes imposed by the increased GHG emission, particularly in the wetter lands around the Caspian and Mediterranean Seas which are simulated as experiencing TWS decrease under global warming. Temporal-ensemble mean TWS due to GHG forcing (Fig. 3b) is only partially reversed by SAI (Fig. 3d), and the water storage shortfall is not fully canceled out by the intervention (Fig., 3c and d). However, simulated TWS in Iran and the southern half of MENA has greater water storage under SAI relative to the historical period (Fig. 3c).

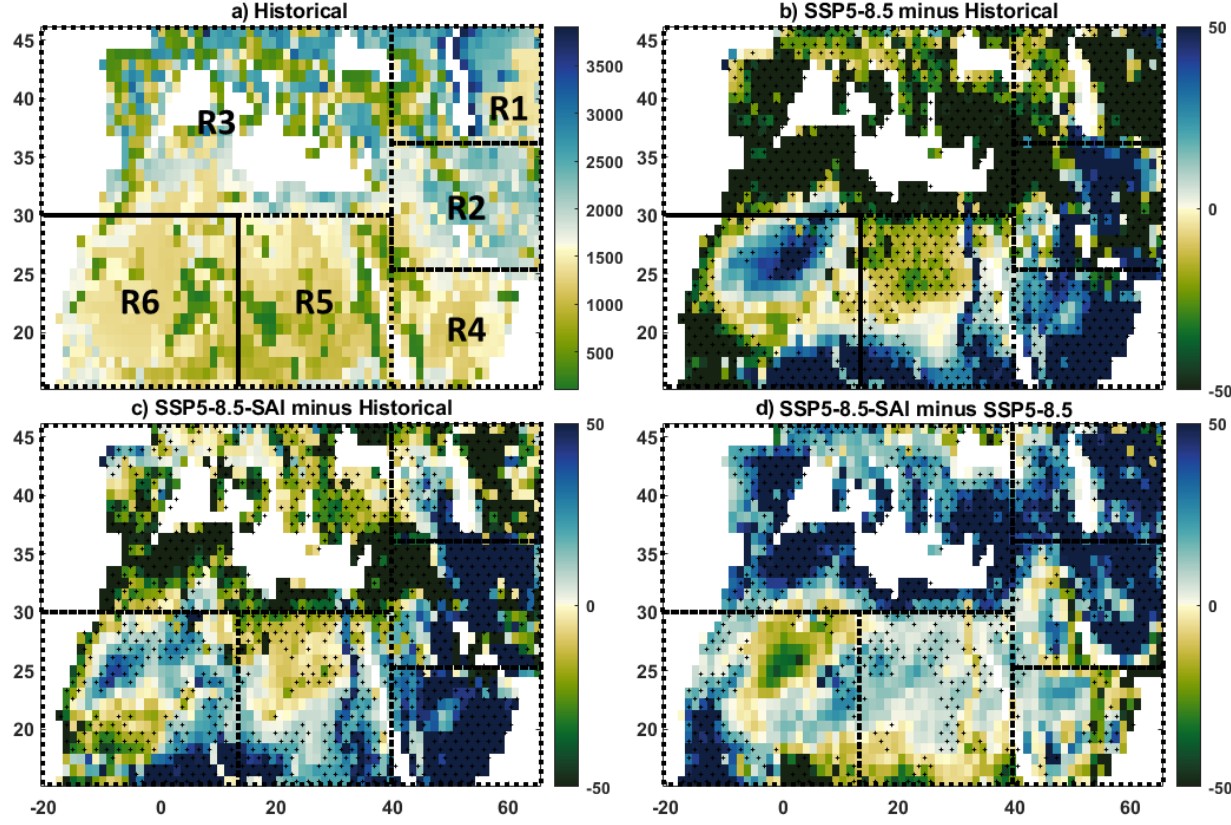

**Figure 3.** Ensemble mean maps of TWS across the studied domain in the historical climate (a) over 1985-2014 and their projected future changes in the 2071–2100 period under the SSP5-85 GHG scenario (SSP5-8.5 minus historical (b) and GHG+SAI minus historical (c)). The extent to which the SAI impacts the TWS changes imposed by global warming is further shown (SSP5-8.5-SAI minus SSP5-8.5 (d)). Hatched areas show where all ensemble members agree on the sign of the changes.

In Fig. 4, we compare how simulated TWS statistical distributions vary between scenarios for each region. Mean TWS significantly ($p<0.05$) decreases in the wetter lands around the Caspian (R1) and Mediterranean (R3) Seas to the north (3.7-5.2% on area average) while it significantly increases in the dry region of Arabian Peninsula (5.6%) in response to GHG warming. SAI, on the whole, partially

reverses the projected changes in TWS from increasing GHG concentrations toward its historical
values. Interestingly, SAI overcompensates the TWS changes imposed by the high GHG forcing in Iran
and Iraq (R2) where this region shows no significant change under GHG emissions (Fig. 4b). SAI also
has an amplifying effect in R5 and a slight overcompensation in R6, but its impact is statistically
insignificant.


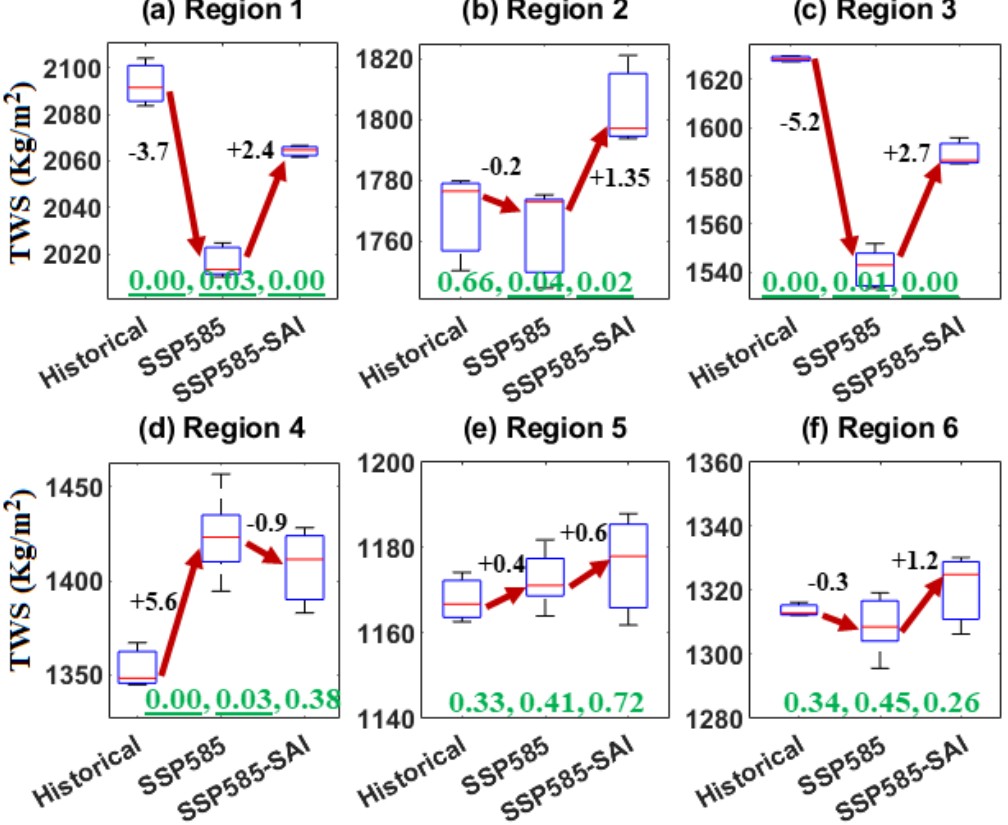

**Figure 4.** Box and whiskers plot of the changes in the Terrestrial Water Storage (TWS) in regions 1
to 6 over 2071-2100 under SSP5-8.5 and SSP5-8.5-SAI relative to historical conditions (1985-
2014). The titles of subplots refer to the regions. The median for each experiment is denoted by the
red line, the upper (75th) and lower (25th) quartiles by the top and bottom of the box, and ensemble
limits by the whisker extents. The positive/negative values in black are the change percent under
SSP5-8.5 and SSP5-8.5-SAI relative to the median of the historical period data. The three values in
green refer to $p$-values between historical and global warming, historical and SAI, and global
warming and SAI, respectively, obtained from $t$-test analysis in which the underlined $p$-values are
statistically significant.


We also compared the changes in TWS with changes in precipitation, temperature, real ET, soil
moisture, and potential ET over each region under both global warming and SAI scenarios (Figs. S7
to S10 in the Supplementary Information). The TWS decreasing patterns under both SSP5-8.5 and
SSP5-8.5-SAI scenarios across the entire study area are similar to soil moisture change patterns (Figs.
S7 and S9 in Supplementary Information), but are more widespread than precipitation under global
warming (Fig. S9). Notably, in the Mediterranean and the dry MENA region, the soil moisture
variability accounts for the dominant component of TWS variability (Pokhrel et al., 2021). However,
the decreased TWS is seen beyond the regions of reduced precipitation (Fig. S9), from beyond the
Mediterranean and Atlantic coasts to include Syria, Iraq, and the lands around the Caspian Sea as well
as to a wide portion of NA (Fig. 3). These include places where precipitation is either increasing or
shows no significant change, consistent with results reported by Cook et al. (2020).

In Summary, our findings show that the SSP5-8.5-SAI scenario has a potential to partially offset the
significant changes in mean TWS imposed by SSP5-8.5 over the entire MENA. While SAI (Fig. 3d)
succeeded in reversing mean TWS deficits in the wetter lands around the Caspian and Mediterranean
Seas driven by the GHG SSP5-8.5 scenario (Fig. 3b), it did not fully cancel out the TWS deficits (Figs.
3c, 4a, and 4c). However, in the dry MENA regions (Fig. 3d), particularly Iran (containing the Lut
desert in the south-east region and the Kavir desert in the north-central), Iraq, and the Arabian
Peninsula (housing the Arabian Desert), SAI resulted in higher mean water storage relative to the
historical period (Figs. 3c and 4).

**3.2 Changes in extreme TWS**
We compared changes in the expected return frequency of comparatively rare events to those during
the historical period. Changes in mean conditions discussed so far are clear, but the changes in
extremes display even larger separations between those expected under pure GHG forcing and the
GHG+SAI scenarios. An increase in the return level or decrease in the return period of TWS means
that the rare levels of high water availability increase, while a decrease in return level for a given
period means that rich water availability events become rarer. We applied a GEV distribution to the
complete dataset of monthly TWS values without explicitly setting maximum values in Fig. 5. For
comparison, we also extracted the annual maximum TWS values and provided the corresponding fitted
GEV distribution. Overall, the probability densities for both datasets exhibit a high degree of similarity
across various regions and scenarios (e.g., Figs. S11 and S12). Additionally, the graphs depicting return
levels versus return periods based on annual maximums (Fig. S13) closely resemble the results obtained
from the entire dataset (Fig. 5). In all cases, the trends are highly similar (compare Figs. 5 and S13),
although it's worth noting that the annual maximums scenario exhibits slightly wider upper and lower
bounds compared to the entire dataset scenario. We therefore focused on the results obtained from the
entire dataset. Fig. 5 shows the return levels versus return period curves with the 95% lower and
upper bands. To determine which curves (including its upper and lower bounds) are significantly
different from each other (*p*-values less than 0.05), we first conducted the repeated measures
analysis of variance which compares means across one or more variables that are based on repeated
observations, and then performed post hoc Tukey-Kramer comparisons. The expected return levels
versus return period curves (Fig. 5) decrease in response to both GHG warming and GHG+SAI in the
Caspian and Mediterranean Seas area (R1 and R3) as well as in the eastern NA (R5) as a more
continental dry land but increase in the Arabian Peninsula (R4) and western NA (R6). In Iran and
Iraq (R2), SAI leads to a significant increase in expected TWS return levels relative to both historical
conditions and the high GHG emission scenarios (Fig. 5b) while SAI tends to partially counteract the
GHG-driven TWS changes in R1, R3, R4, and R5. Larger TWS levels are expected for the entire MENA
compared with the GHG climate alone, particularly in Iran, Iraq, and the western NA. Nonetheless,
compared to the historical period, the Arabian Peninsula (Fig. 5d) is the region with the most robust
increase in the extreme TWS under both the global warming and SAI scenarios. Extreme TWS in its
neighbor dry land of eastern NA with a hyper-arid climate is still smaller than the historical
conditions.

Table 2 quantitatively compares the differences between TWS (and its corresponding 95% lower and
upper bounds in Fig. 5) changes at 30-, 50-, and 100-yr return periods under historical, global
warming, and SAI scenarios. Global warming, on the whole, decreases the TWS extremes (i.e., fewer
wetter conditions) at 30- to 100-year return periods over all the study areas except for the Arabian
Peninsula (R4) and western NA (R6). The most robust decreases in the extreme TWS imposed by
global warming relative to historical conditions occur in the lands around the Caspian R1 (-108% on
average over return periods from 30- to 100-year) and Mediterranean R3 (-43% on average) and the
eastern NA R5 (-89% on average) are partially suppressed by SAI. A small increase in the extreme
TWS in Iran and Iraq (R2) simulated under GHG (+15%) is overcompensated by SAI (+65%).
Although SAI decreases the TWS in the Arabian Peninsula (-11%) relative to global warming, it still
tends to experience the most robust extreme water storage increases in the future (+153%)
compared with historical conditions. In western NA (R6), the SAI simulation slightly intensifies the
increased extreme TWS imposed by high GHG emissions by +27%. Although SAI partially
compensates for the changes over most of the study area (positive SSP5-8.5-SAI minus SSP5-8.5
values in Table 2), on the whole, extreme TWS tend to increase in the dry regions of Iran and Iraq,
the Arabian Peninsula, and western NA while substantially decreasing in the wetter lands around the
Caspian and Mediterranean Seas, and to lower degrees, in the eastern NA as a more continental dry
land compared with historical conditions.

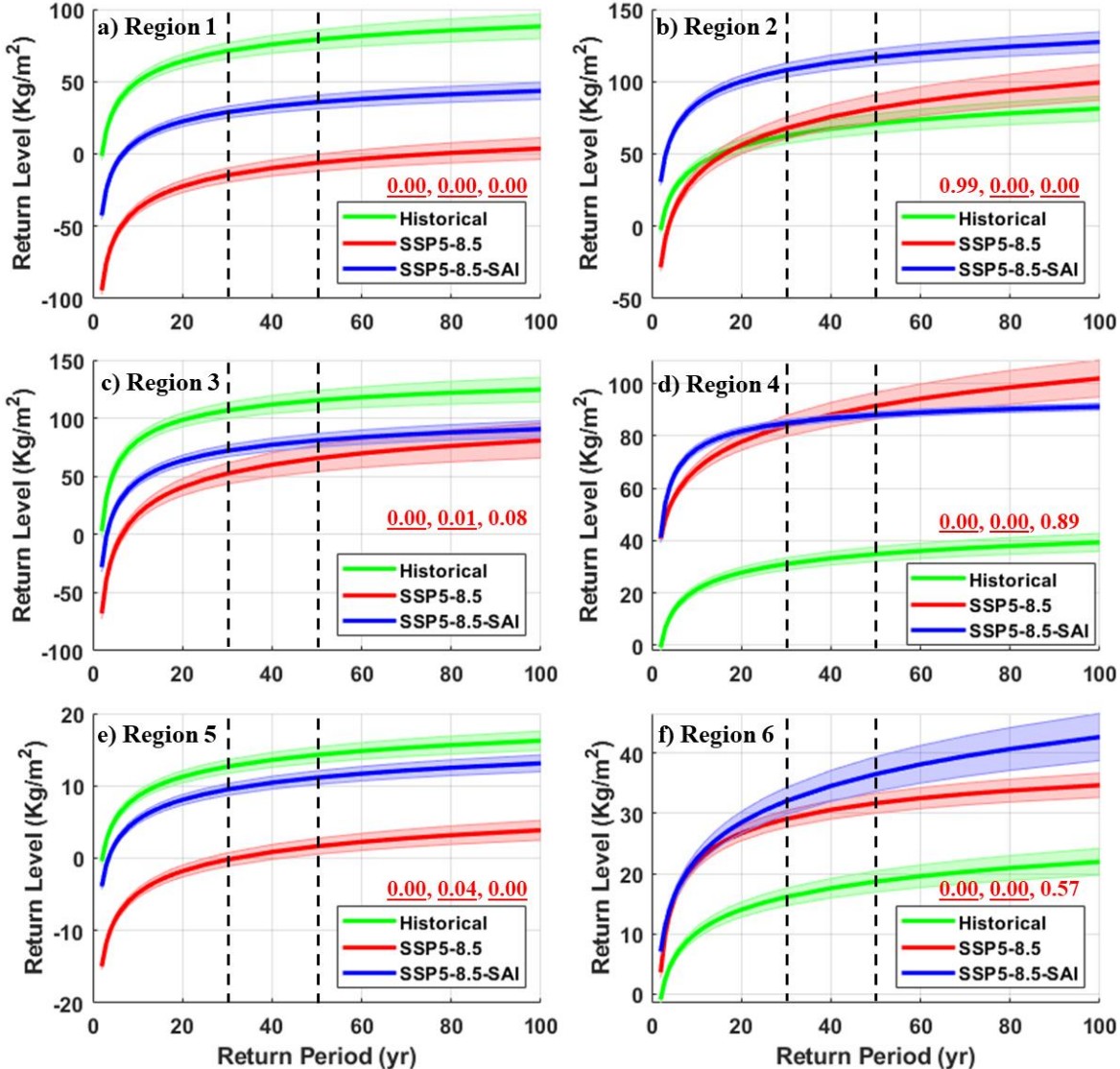


**Figure 5.** The TWS anomaly return level versus return period using the first three realizations for the historical, SSP5-8.5, and SSP5-8.5-SAI in regions 1 to 6 (a to f). The two parallel dashed black lines refer to 30- (left) and 50-year (right) return periods. Shading in each curve is the 95% upper and lower confidence bands. The three values in red refer to $p$-values between historical and global warming, historical and SAI, and global warming and SAI, respectively, obtained from the repeated measures analysis of variance and the post hoc Tukey-Kramer comparisons in which the underlined $p$-values are statistically significant.



**Table 2.** The percent differences (%) between the medians of the TWS return level at 30-, 50-, and 100-year return periods using the first
three realizations for the historical, SSP5-8.5, and SSP5-8.5-SAI. Consistently, the value inside the parenthesis is the percent difference-
range values between lowers and uppers 95% confidence intervals from different scenarios.

| | (SSP5-8.5 – Historical)/Historical*100 | | | (SSP5-8.5-SAI – Historical)/ Historical*100 | | | (SSP5-8.5-SAI – SSP5-8.5)/ Historical*100 | | |
|---|---|---|---|---|---|---|---|---|---|
| Region | 30-yr | 50-yr | 100-yr | 30-yr | 50-yr | 100-yr | 30-yr | 50-yr | 100-yr |
| R1 | −121 (-130, -113) | -108 (-117, -100) | -96 (-105, -88) | −59 (-62, -57) | -55 (-57, -53) | -51 (-53, -49) | 61 (56, 68) | 53 (48, 60) | 45 (40, 52) |
| R2 | 8 (6, 11) | 15 (12, 17) | 22 (20, 24) | 73 (66, 81) | 65 (58, 73) | 57 (50, 65) | 64 (55, 75) | 50 (41, 60) | 34 (25, 46) |
| R3 | -51 (-56, -46) | -43 (-49, -38) | -35 (-42, -29) | -33 (-34, -32) | -30 (-31, -29) | -27 (-28, -26) | 18 (14, 24) | 13 (8, 20) | 8 (2, 16) |
| R4 | 170 (163, 178) | 163 (157, 169) | 160 (155, 164) | 173 (158, 191) | 153 (138, 170) | 132 (117, 150) | 4 (-4, 13) | -10 (-19, 1) | -27 (-39, -14) |
| R5 | -102 (-110, -95) | -89 (-96, -82) | -76 (-83, -70) | -25 (-26, -24) | -22 (-23, -21) | -19 (-20, -18) | 77 (70, 84) | 67 (61, 73) | 57 (52, 63) |
| R6 | 80 (73, 89) | 70 (63, 77) | 58 (52, 65) | 99 (95, 103) | 95 (93, 99) | 94 (93, 96) | 18 (14, 22) | 26 (21, 30) | 36 (31, 41) |



**3.3 Drivers of TWS change**

To assess which variables have the most impact on mean TWS under both global warming and SAI, we fitted an MLR model to each ensemble member separately in each of the six regions (Figs. 6 and 7). The most important variable for the mean TWS under both global warming and SAI scenarios is region-specific. In the wet lands surrounding the Caspian (R1) and Mediterranean (R3) Seas, temperature and precipitation are the primary drivers of TWS changes. In contrast, in the Middle East, characterized by predominantly dry climates (R2 and R4), vegetation coverage (i.e., LAI) plays a dominant role. This observation aligns with the fact that temperature limits ET in the wet regions, while in arid and hot regions, the availability of water for ET is the predominant limiting factor (Bao et al., 2021). In NA, where TWS changes are irregular (Fig. 2), temperature holds the greatest significance in the eastern regions (R5), while real ET is the primary driver in the west (R6). Warmer climate enhances the atmospheric water content over regions and seasons (Cook et al., 2020) since 1°C warming is accompanied by ~7% enhancement in the air water storage capacity (Trenberth, 2011), and, in turn, increases the evaporative demand (Arnell, 1999), and vice versa for cooler conditions. Real ET itself is mostly controlled by temperature and available water for evaporation (i.e., precipitation, soil moisture, and vegetation coverage). With just temperature and precipitation as independent variables, we find that the temperature under both global warming and SAI is generally more important for TWS than precipitation over the wet lands around the Caspian and Mediterranean Seas as well as the eastern NA. In contrast, precipitation plays a stronger role on TWS in Iran, Iraq, and the western NA with lower precipitation under both future climate scenarios.

The regression models indicate that TWS is mostly driven by the combined impacts of changes in vegetation coverage, real ET, temperature and precipitation, consistent with the fact that precipitation is not the only controlling factor for water resources (Cook et al., 2014; Wu et al., 2020). However, the temperature in the Mediterranean area with the highest precipitation over the entire domain studied plays a more important role than precipitation, vegetation coverage, and real ET under both warming and SAI scenarios.

Caution is required when interpreting the relative importance results for the arid regions of R4 to R6 as their variance explained ($R^2$=0.3 to 0.52) from the MLR models is smaller than those (up to 0.89 and 0.96) for the wetter lands around the Caspian and Mediterranean Seas. This, most probably, arises from the arid to hyper-arid climate of R4 to R6 with a small and irregular annual precipitation, and, in turn, irregular TWS anomaly time series (Figs. 2d, e, and f).

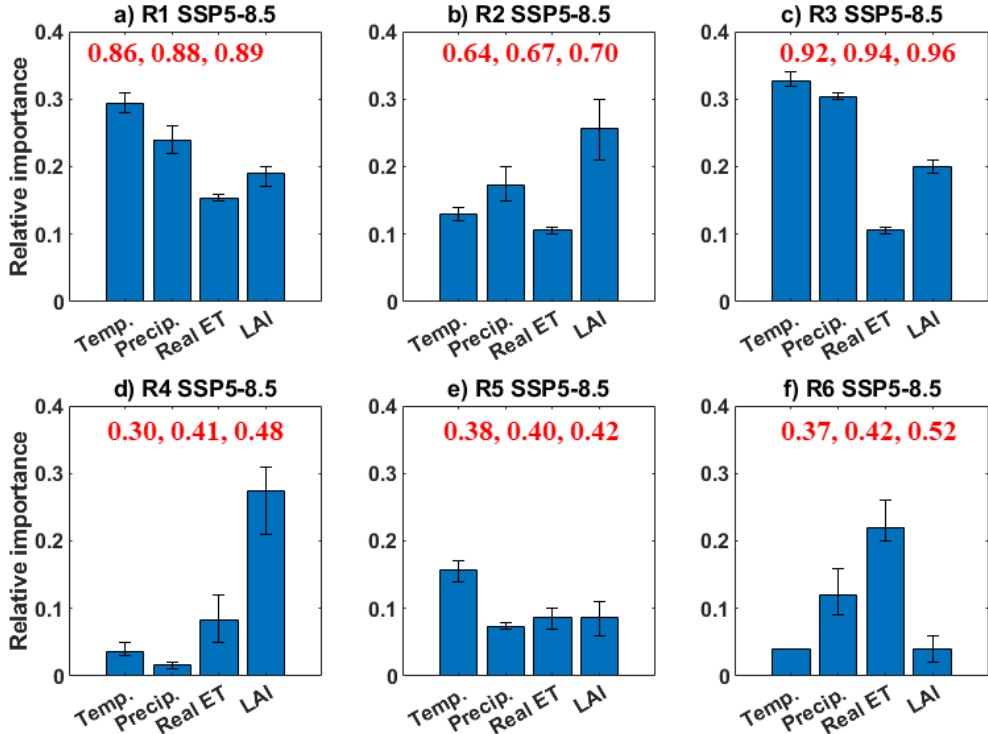

**Figure 6.** LMG importance plot (Lindeman et al., 1980) of the four independent variables in the
regression for TWS for the global warming SSP5-8.5 scenario in each region. The bar and range-bar
respectively show the ensemble mean importance and the range of importance from the three
ensemble members. The three values in red on each subplot shows the minimum, mean, and

maximum variances explained by models.


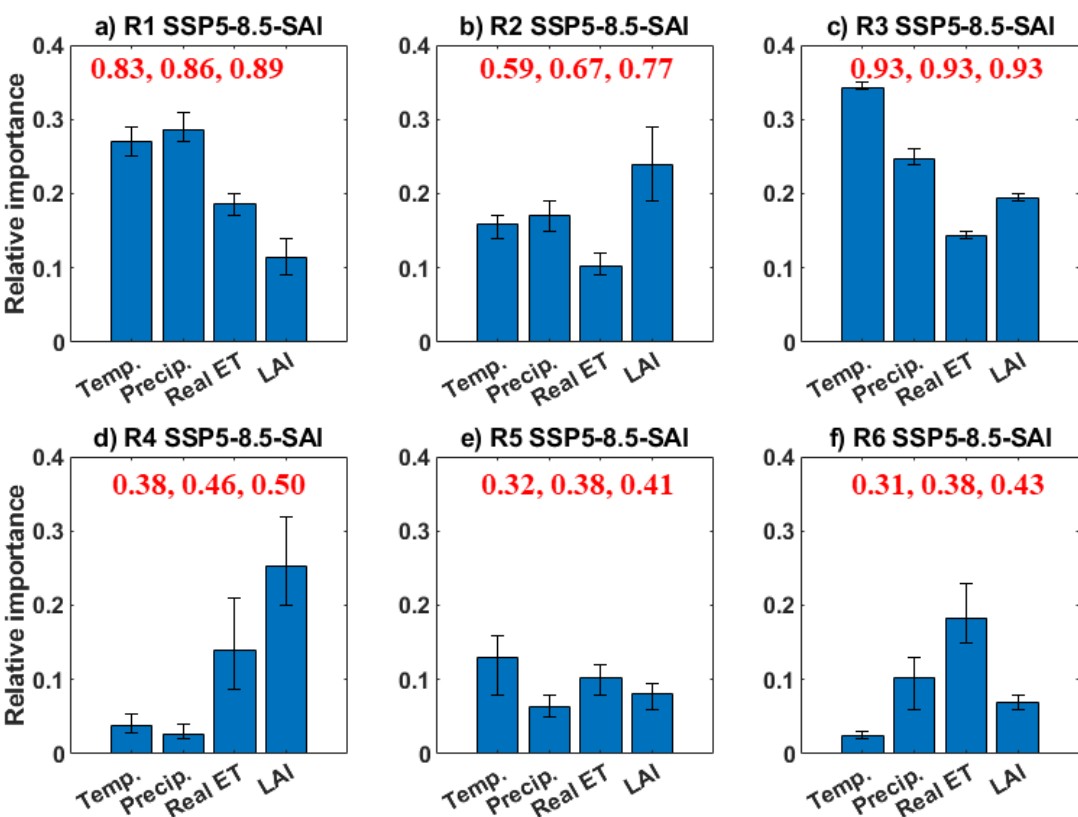

**Figure 7.** As in Fig. 6, but for the SSP5-8.5-SAI scenario.

## 4. Discussion

We have analyzed the potential impacts of the unmitigated global warming SSP5-8.5 scenario (GHG) and the same GHG emissions trajectory with the addition of SAI (GHG+SAI) on both the mean and extreme water storage across the lands around the Caspian and Mediterranean Seas, Middle East, and NA. We have used the CESM2(WACCM) climate model simulations with three realizations of each historic and SSP5-8.5-SAI scenario and five available realizations for SSP5-8.5. In response to high GHG emission over the 2071-2100 period, the mean TWS decreases in the wetter regions (i.e., around the Caspian and Mediterranean Seas with mild wet winters and warm to hot, dry summers), in agreement with the previous studies based on SSP5-8.5 (e.g., Cook et al., 2020; Scanlon et al., 2023), RCP2.6 and RCP4.5 (e.g., Döll et al., 2018) as well as with projections from 11 global hydrological models (Schewe et al., 2014) with globally forced 2°C warming (Schleussner et al., 2016). Similarly, a decrease in precipitation (Kim and Byun, 2009), surface runoff (Cook et al., 2020), and TWS (Pokhrel et al., 2021) has been reported across Mediterranean coasts under GHG warming. In contrast, the mean TWS increases or shows no significant change in the MENA, housing several major deserts with minimal precipitation. The temporal-ensemble mean TWS increase in the southern

MENA is consistent with other climate model simulations showing increased precipitation and soil
moisture in CMIP6 simulations under SSP5-8.5 (Cook et al., 2020), and SSP2-4.5 (Ajjur et al., 2021;
Scanlon et al., 2023). This further aligns with a projected northward shift of the inter-tropical
convergence zone (ITCZ) in eastern Africa, mostly during a months of May to October (Mamalakis et
al., 2020), leading to increased moisture transfer to the Southern Middle East and NA (Waha et al.,

2017).


Given the prevailing water scarcity challenges in many regions of the Middle East where population
growth is a continuing concern (Oroud, 2008), by mitigating the vulnerability to global warming, SAI
may offer a potential strategy to augment the regional water resources across the area, particularly
in the dry regions of Iran (containing the Lut desert in the south-east region and the Kavir desert in
the north-central), Iraq, and the Arabian Peninsula (housing the Arabian Desert), as compared with
the pure GHG forced scenario. Similarity, Jones et al. (2018) found that SAI could effectively
counteract the changes in available water imposed by global warming on Earth's lands. Mousavi et
al. (2023) also found increased soil moisture and enhanced vegetation coverage would lead to the
reduction of dust concentration in the MEAN region under SAI.

The more robust and widespread deficit in mean TWS compared to precipitation in the area, which
is in line with results reported by Cook et al. (2020), highlights the profound roles that other
variables/processes have on the increased ET such as greater atmospheric moisture demand (Dai et
al., 2013, 2018) and greater vegetation water use (Mankin et al., 2019) owing to warmer conditions
under global warming, consistent with regression model results. According to MLR model results
(Figs. 6 and 7), the projected changes in TWS were not solely attributable to precipitation; its
interplay with other factors, such as vegetation coverage, temperature, and ET play a pivotal role.
The vegetation coverage as the primary variable influencing changes in TWS in the MENA region
substantially increases under global warming (Figs. S14 and S15). It has an important, but often
complex and uncertain, role in surface water content (Lemordant et al., 2018; Trugman et al., 2018);
the denser vegetation coverage, the higher evapotranspiration rates. Furthermore, although
precipitation over a broad portion of MENA is lowered under SAI relative to global warming, the
mean TWS, in general, increases across a broad portion of the MENA region in response to the
intervention. TWS significantly increases over Iran and Iraq under SAI compared to historical and
global warming (Fig. 4b) as gains in available water from decreased temperature and, in turn, ET is
largely sufficient to compensate for decreased precipitation (Figs. S8 and S10), signifying that in
addition to precipitation, the water storage also strongly depends on local temperature (Ajjur et al.,
2021). As an example, around the Caspian Sea (R1), although the changes in precipitation imposed
by global warming are simulated to have been fully restored by SAI, the temperature has not; and in
turn, the TWS is not fully restored by SAI. This is consistent with MLR model results (Fig. 7a) in which,
beyond the precipitation, temperature also plays an important role in TWS across R1. Other studies
also found that changes in precipitation does not necessarily correlate with changes in surface water,
due to differences in precipitation and evaporation responses under SAI (Irvine et al., 2016).

Our findings, on the whole, suggest that the specific SAI scenario considered here could help water
storage in the dry regions (R2, R4, R5, and R6), i.e., leads to higher soil moisture and TWS compared
with both the historical conditions and pure GHG-induced global warming. Likewise, Dagon and
Scharg (2017) documented a rise in mean water availability and soil moisture during a period of June
to August in MENA using SolarGeo simulations, consistent with the significant reduction in daily
maximum temperatures and ET across the Middle East. This works through the combined positive
effects of (1) a substantial decrease in temperature and ET over the entire study area compared with
SSP5-8.5 global warming, and (2) the increased precipitation in the southern MENA dry regions
relative to historical conditions. The Middle East may therefore benefit from the water enrichment
from climate change through the implementation of solar intervention (Burnell, 2021). However, the
wet and colder regions, particularly around the Mediterranean coasts, may have less water storage
compared with the historical period but more water relative to the GHG scenario due to a significant
decrease in ET under SAI. Simpson et al. (2019) also reported a noteworthy decline of 18.5% in
available water (precipitation minus evaporation) across the Mediterranean area under high GHG
emissions while it has been partially reversed (only 5%) by a decrease in evaporation under SAI.

Although SAI partially compensates for the extreme TWS changes in most of the study area, aligning
with findings by Jones et al. (2018), the overall extreme TWS trend indicates an increase in dry
regions of Iran and Iraq, Arabian Peninsula, and western NA. Conversely, there is a substantial
decrease in extreme TWS in the wetter lands around the Caspian and Mediterranean Seas, and to
lower degrees, in the eastern NA compared to historical conditions. The implications of our findings
under both future climate scenarios (SSP5-8.5 and SSP5-8.5-SAI) extend beyond hydrology and
water resources management. Changes in TWS have significant implications for climate adaptation,
flood and drought risk management, and infrastructure planning. Some dry areas such as Iran, Iraq,
and the Arabian Peninsula are projected to receive greater extreme TWS under both global warming
and SAI or only SAI, and these regions have suffered historically from flooding (e.g., Abbaspour et al.,
2009; Ghavidel and Jafari Hombari, 2020; Dezfuli et al., 2021). The significant increase in extreme
TWS enhances their flood risks. Hence, governments in these regions should plan for adaptations to
water megastructures such as the dams on the large rivers of Karkheh and Karun in western Iran and
the Euphrates and Tigris in Iraq, since they have been mostly designed with historical hydrology in
mind.

There are several caveats and caution needed for our results. First, our findings are based on a single
model simulation (CESM2) and a single scenario climate scenario SSP5-8.5 with (three available
realizations) and without (five available realizations) SAI. Future studies should consider alternative
SAI scenarios to explore the sensitivity of our results to model and scenario choices. The SSP
scenarios include some that clearly portray undesirable futures, especially the high emissions SSP5
scenarios or the regional rivalry SSP3 that illustrate the danger of unchecked climate change
(MacMartin et al., 2022). There are more caveats for the SAI experiment used here (1) it deploys in
2020, therefore does not simulate any plausible future, and (2) takes into account solely the high-
emissions scenario SSP5-8.5 that is suitable for capturing a high "signal" compared to internal
variability. This is useful for understanding the science but inconsistent with present-day projections
of mitigation attempts (Burgess et al., 2020). However, while the signal is stronger under high GHG
emissions, it is plausible that the directions and patterns of response would be similar in a lower-
emission experiment, with the magnitude of changes roughly depending on the degree of warming
being suppressed by SAI (e.g., MacMartin et al., 2022).

**5. Conclusions**

The current study is the first attempt for understanding the influence of GHG emission and SAI
scenarios on both mean and extreme water storage changes over the lands around the Caspian and
Mediterranean Seas, Middle East, and northern Africa under global warming and SAI scenarios
compared to the historical 1985-2014 conditions. The mean TWS is projected to decrease across the
wetter lands around the Caspian and Mediterranean Seas to the north (3.7-5.2% on average) but
increase over the most MENA region (up to 5.6% over the Arabian Peninsula) that has a drier climate
under the high GHG forcing compared to the present-day conditions.

Although the SAI tends to reverse, to a degree, the significant changes in TWS revealed by SSP5-8.5
over the entire area, it significantly overcompensates for the slightly reduced TWS under the high
GHG scenario in Iran and Iraq. MLR model analysis of driving factors suggests that the impacts of
temperature on water storage changes, like precipitation, are also important under both high GHG
forcing and SAI scenarios. Although SAI mostly decreases precipitation over most of the domain, it is
accompanied by higher mean TWS across the entire study area due to the cooler climate.

Although significant changes in the extreme TWS under high GHG emissions are reduced by SAI, the
changes due to both future climate changes are still large relative to the historical period across a
broad portion of the domain. With SAI, TWS significantly decreases in the eastern lands around the
Caspian Sea while substantially increasing across the Middle East regions of Iran, Iraq, and the
Arabian Peninsula. This may increase flood risks since water megastructures have been mostly
designed with historical hydrology in mind. Finally, the SAI scenario appears to increase accessible
water storage in the dry regions of the Middle East and northern Africa. The wetter and colder lands
around the Caspian and Mediterranean Seas may have less available water compared with the
historical conditions, although SAI partially ameliorates the changes imposed by global warming.

**Data availability:**
The data for CESM2 simulations are publicly available via its website: https://esgf-
node.llnl.gov/search/cmip6/. To access these specific data via ESGF website use the Source ID =
CESM2-WACCM, Experiment ID=ssp585, and Frequency = mon. The SSP5-8.5-SAI data are freely
available at https://www.earthsystemgrid.org/dataset/ucar.cgd.ccsm4.geomip.ssp5.html
(https://doi.org/10.26024/t49k-1016).

**Acknowledgments:**
We appreciate the financial support from the DEGREES Initiative in collaboration with the World
Academy of Sciences (TWAS) under grant no. 4500443035.

**Author contributions:**
AR: coordinated the analysis, graphics of various figures, and manuscript preparation; KK and ST:
conceptualized and prepared the data; and JCM: conceptualized and coordinated the interpretation
and discussion for various sections. All authors contributed to the discussion and writing.

**Competing the Interest:**
The contact author has declared that none of the authors has any competing interests.

**Financial support:**

This research has been supported by the DEGREES Initiative in collaboration with the World Academy of Sciences (grant no. 4500443035).

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
