# Peer review of "Future water storage changes over the Mediterranean, Middle East, and North Africa in"

_EGUsphere, 2023_

## Author Comment (AC1)

**Response to RC1:**

Rezaei et al. use climate model simulations to investigate the impacts of stratospheric aerosol injection (SAI) on the Middle East and North African (MENA) region. The study looks specifically at total water storage, and associated hydrology variables, in model simulations with SAI and climate change. MENA is an understudied region in the context of SAI, with important potential impacts on the water cycle, so this paper is a welcome addition to the literature. The paper overall needs reorganization and edits to the text for clarity, as well as modifications to the figures and explanation of statistical methods to better communicate the results. General comments are listed first, and further specific comments below.

Reply: We sincerely appreciate your effort and time in detailed reviewing our manuscript as well as constructive comments/suggestions. We have attempted to revise the manuscript in the light of your suggestions/comments. The following is also the point–point response to all the comments (comments are rewritten in black color and their corresponding replies in red).

**General comments:**

- I enjoyed reading about the study area in section 2.1 (in particular, the first two paragraphs). It would be informative to return to this context in the discussion section and talk about how the results in this paper might impact the regional climate more broadly.

Reply: Agreed. We have done this now in several places in the discussion, in particular with new text below.

"In response to high GHG emissions over the 2071-2100 period, TWS decreases in the wetter regions (i.e., around the Caspian and Mediterranean Seas with mild wet winters and warm to hot, dry summers) while, on the whole, it increases or shows no significant change in the MENA, housing several major deserts with minimal precipitation."

"Given the prevailing water scarcity challenges in many regions of the Middle East, SAI may offer a potential strategy to augment the regional water resources across the area, particularly in the dry regions of Iran (containing the Lut desert in the central region and the Kavir desert in the east), Iraq, and the Arabian Peninsula (housing the Arabian Desert) as compared with the pure GHG forced scenario."

"The MENA region contains several deserts such as the Sahara in NA, the Arabian in the Arabian Peninsula, and the Lut and Kavir in Iran. As a result, it features several dust hotspots (Mousavi et al., 2023). One noteworthy aspect of our findings is the projection of higher water storage, increased soil moisture, and enhanced vegetation coverage under the solar geoengineering (SAI) scenario relative to historical conditions. This evidence holds the promise of potentially reducing dust concentrations in MENA in the future, in line with reduced dust concentrations projections made by Mousavi et al. (2023) using the GLENS project."

It would be helpful to give additional context for why these specific CESM simulations were used here (e.g., instead of the large ensemble simulations like GLENS/ARISE). I think a novelty of Tilmes et al. 2020 was the overshoot scenario but I don't believe those simulations were used here. I suggest adding some additional text in section 2.2 for context on the model simulations.

Reply: We have added the following additional text to section 2.2:

"CESM2 ranks among the top nine models known for their accuracy in simulating global precipitation patterns, based on the Hellinger distance metric, which compares the bivariate empirical densities of CESM2 with those of 34 CMIP6 models, against historical precipitation data sourced from the Global Precipitation Climatology Centre (GPCC) (Abdelmoaty et al., 2021). CESM2 has precipitation biases about 20% lower than CESM1 (Danabasoglu et al., 2020). CESM2(WACCM6) has improved stratospheric aerosol treatment (Danabasoglu et al., 2020) that is much more consistent with observations than previous climate models (Mills et al., 2016). Furthermore, CESM2(WACCM), including improvements in the simulation of carbon and nitrogen cycles with global land carbon trends matching observations remarkably well (Danabasoglu et al., 2020), and a full ocean model (Parallel Ocean Program version 2, POP2) with modularized biogeochemistry (Marine Biogeochemistry Library, MARBL) to more fully simulate the stratospheric response to the high GHG warming in the SSP5-8.5 scenario (Koven et al., 2022)."

Ref:
Abdelmoaty, H. M., Papalexiou, S. M., Rajulapati, C. R., & AghaKouchak, A. (2021). Biases beyond the mean in CMIP6 extreme precipitation: A global investigation, Earths Future, 9, e2021EF002196.
Danabasoglu, G., Lamarque, J. F., Bacmeister, J., Bailey, D. A., DuVivier, A. K., Edwards, J., ... & Strand, W. G. (2020). The community earth system model version 2 (CESM2). Journal of Advances in Modeling Earth Systems, 12(2), e2019MS001916.
Koven, C. D., Arora, V. K., Cadule, P., Fisher, R. A., Jones, C. D., Lawrence, D. M., ... & Zickfeld, K. (2022). Multi-century dynamics of the climate and carbon cycle under both high and net negative emissions scenarios. Earth System Dynamics, 13(2), 885-909.

Generally, the paper does a good job specifying which comparison is being made (e.g., SSP relative to historical or SAI relative to historical) but there are some additional places to clarify the text in the results and discussion sections, which will make these sections easier to follow and interpret. I have some specific suggestions in the comments below.

The discussion section currently recaps/repeats many of the results (e.g., Lines 485-507). I suggest focusing on interpreting the results, highlighting particularly interesting results, and connecting with previous studies. Some of this is already present in the discussion section (e.g., Lines 438-442, 475-483, 509-518), so I think the section just needs some edits to move results into the results section and focus the discussion.

Reply: We have greatly revised the discussion. We moved the following paragraphs into the results (Section 3.1):

"We also compared the changes in TWS with changes in precipitation, temperature, real ET, soil moisture, and potential ET over each region under both global warming and SAI scenarios (Figs. S2 to S6 in the Supplementary Information). The TWS decreasing patterns under both SSP5-8.5 and SSP5-8.5-SAI scenarios across the entire study area are similar to soil moisture change patterns (Fig. S2 and S4 in Supplementary Information) but more widespread than precipitation under global warming (Fig. S4). The decreased TWS is seen beyond the regions of reduced precipitation (Fig. S4), from beyond the Mediterranean and Atlantic coasts to include Syria, Iraq, and the lands around the Caspian Sea as well as to a wide portion of NA (Fig. 4). These include

places where precipitation is either increasing or shows no significant change, consistent with results reported by Cook et al. (2020).

In Summary, our findings show that the SSP5-8.5-SAI scenario has potential to partially offset the significant changes in mean TWS imposed by SSP5-8.5 over the entire MENA. While SAI (Fig. 3d) succeeded in reversing mean TWS deficits in the wetter lands around the Caspian and Mediterranean Seas driven by the GHG SSP5-8.5 scenario (Fig. 3b), it did not fully cancel out the TWS deficits (Figs., 3c, 4a, and 4c). However, in the dry MENA regions (Fig. 3d), particularly Iran (containing the Lut desert in the central region and the Kavir desert in the east), Iraq, and the Arabian Peninsula (housing the Arabian Desert), SAI resulted in higher mean water storage relative to the historical period (Figs. 3c and 4)."

In addition, we also revised following parts of the discussion:

"In response to high GHG emission over the 2071-2100 period, the mean TWS decreases in the wetter regions (i.e., around the Caspian and Mediterranean Seas with mild wet winters and warm to hot, dry summers) while, on the whole, it increases or shows no significant change in the MENA, housing several major deserts with minimal precipitation. Similarly, a decrease in precipitation (Kim and Byun, 2009), surface runoff (Cook et al., 2020), and TWS (Pokhrel et al., 2021) has been reported across Mediterranean coasts under GHG warming. The mean TWS increase in the southern MENA is consistent with other climate model simulations showing increased precipitation and soil moisture in CMIP6 simulations under SSP5-8.5 (Cook et al., 2020), and SSP2-4.5 (Ajjur et al., 2021; Scanlon et al., 2023). This further aligns with a projected northward shift of the inter-tropical convergence zone (ITCZ), leading to increased moisture transfer to the Southern Middle East and North Africa (Waha et al., 2017). The CMIP6 models have consistently projected a substantial northward shift of the ITCZ in eastern Africa, mostly during the months of May to October (Mamalakis et al., 2021)."

"By mitigating the vulnerability to global warming, particularly in MENA where population growth is a continuing concern (Oroud, 2008), SAI interventions could play a crucial role in water resource management and adaptation strategies. Similarity, Jones et al. (2018) found that SAI could effectively counteract the changes in available water imposed by global warming on Earth's lands."

"The MENA region contains several deserts such as the Sahara in NA, the Arabian in the Arabian Peninsula, and the Lut and Kavir in Iran. As a result, it features several dust hotspots (Mousavi et al., 2023). One noteworthy aspect of our findings is the projection of higher water storage, increased soil moisture, and enhanced vegetation coverage under the solar geoengineering (SAI) scenario relative to historical conditions. This evidence holds the promise of potentially reducing dust concentrations in MENA in the future, in line with reduced dust concentrations projections made by Mousavi et al. (2023) using the GLENS project"

"The more robust and widespread deficit in mean TWS compared to precipitation in the area, which is in line with results reported by Cook et al. (2020), highlights the profound roles that other variables/processes have on the increased ET such as greater atmospheric moisture

demand (Dai et al., 2013, 2018) and greater vegetation water use (Mankin et al., 2019) owing to warmer conditions under global warming, consistent with regression model results."

"The vegetation coverage as the primary variable influencing changes in TWS within the area (except for the eastern NA) substantially increases under global warming (Figures S6 and S7). It has an important, but often complex and uncertain, role in surface water content (Lemordant et al., 2018; Trugman et al., 2018); the denser vegetation coverage, the higher evapotranspiration rates."

"This is consistent with MLR model results (Fig. 7a) in which, beyond the precipitation, temperature also plays an important role in TWS across R1. Similarly, while precipitation on lands decreases under SAI relative to global warming, the evaporation decreases are stronger, resulting in a net increase in surface water (Irvine et al., 2016). These regional nuances underscore the necessity of tailored water management strategies to address the complex interplay of these variables."

"Our findings, on the whole, suggest that the specific SAI scenario considered here helps water storage in the dry regions (R2, R4, R5, and R6), i.e., leads to higher soil moisture and TWS compared with both the historical conditions and pure GHG-induced global warming. Likewise, Dagon and Scharg (2017) documented a rise in mean water availability and soil moisture during the period of June to August in MENA using SolarGeo simulations. This works through the combined positive effects of (1) a substantial decrease in temperature and ET over the entire study area compared with SSP5-8.5 global warming, consistent with the significant reduction in daily maximum temperatures and ET across the Middle East, leading to increased soil water under SAI (Dagon and Scharg, 2017) and (2) the increased precipitation in the southern MENA dry regions relative to historical conditions. The Middle East may therefore benefit from the water enrichment from climate change through the implementation of solar intervention (Burnell, 2021)."

"However, the wet and colder regions, particularly around the Mediterranean coasts, may have less water storage compared with the historical period but more water relative to the GHG scenario due to a significant decrease in evapotranspiration under SAI. Simpson et al. (2019) also reported a noteworthy decline of 18.5% in available water (precipitation minus evaporation) across the Mediterranean area under high GHG emissions while it has been partially reversed (only 5%) by decrease in evaporation imposed by GLENS SAI project."

"Although SAI partially compensates for the extreme TWS changes in most of the study area, aligning with the global findings by Jones et al. (2018), the overall extreme TWS trend indicates an increase in dry regions of Iran and Iraq, Arabian Peninsula, and western NA."

New Ref:
Burnell, L. (2021). Risks to global water resources from geoengineering the climate with solar radiation management (Doctoral dissertation, University of Nottingham).
Dagon, K., & Schrag, D. P. (2017). Regional climate variability under model simulations of solar geoengineering. Journal of Geophysical Research: Atmospheres, 122(22), 12-106.

Irvine, P. J., Kravitz, B., Lawrence, M. G., & Muri, H. (2016). An overview of the Earth system science of solar geoengineering. Wiley Interdisciplinary Reviews: Climate Change, 7(6), 815-833.

Jones, A. C., Hawcroft, M. K., Haywood, J. M., Jones, A., Guo, X., & Moore, J. C. (2018). Regional climate impacts of stabilizing global warming at 1.5 K using solar geoengineering. Earth's Future, 6(2), 230-251.

Lemordant, L., Gentine, P., Swann, A. S., Cook, B. I., & Scheff, J. (2018). Critical impact of vegetation physiology on the continental hydrologic cycle in response to increasing CO2. Proceedings of the National Academy of Sciences, 115(16), 4093. https://doi.org/10.1073/pnas.1720712115.

Mamalakis, A., Randerson, J.T., Yu, J.-Y., Pritchard, M.S., Magnusdottir, G., Smyth, P. et al. (2021) Zonally opposing shifts of the intertropical convergence zone in response to climate change, 45.

Mousavi, S. V., Karami, K., Tilmes, S., Muri, H., Xia, L., and Rezaei, A. (2023). Future dust concentration over the Middle East and North Africa region under global warming and stratospheric aerosol intervention scenarios, Atmos. Chem. Phys., 23, 10677–10695, https://doi.org/10.5194/acp-23-10677-2023

Pokhrel, Y., Felfelani, F., Satoh, Y., Boulange, J., Burek, P., Gädeke, A., ... & Wada, Y. (2021). Global terrestrial water storage and drought severity under climate change. Nature Climate Change, 11(3), 226-233.

Trugman, A. T., Medvigy, D., Mankin, J. S., & Anderegg, W. R. L. (2018). Soil moisture stress as a major driver of carbon cycle uncertainty. Geophysical Research Letters, 45, 6495–6503. https://doi.org/10.1029/2018GL078131

Waha, K., Krummenauer, L., Adams, S., Aich, V., Baarsch, F., Coumou, D. et al. (2017) Climate change impacts in the Middle East and northern Africa (MENA) region and their implications for vulnerable population groups. Regional Environmental Change, 17(6), 1623–1638. Available from: https://doi.org/10.1007/s10113-017-1144-2

**Specific comments:**

Lines 68-72: This paragraph seems out of place. Suggest moving this down to where projected future changes in the Mediterranean are discussed (e.g., Line 96).

Reply: Implemented.

Lines 74-84: Moving this paragraph down to after the discussion on climate change impacts (e.g., Line 108) would then transition to an introduction to SRM and associated impacts.

Reply: Implemented.

Line 90: Are there any more recent modeling studies (e.g., CMIP5 or CMIP6) that discuss projected changes in the MENA region? If not, it is worth pointing that out.

Reply: Here we focused on CMIP6 literature for consistency. Please see the revised version of these section copied below:

"Using GHG emission scenario A1B simulated by nine CMIP3-class climate models, Droogers et al. (2012) projected that 22% of the future annual water shortage, 199 km3 in 2050 in MENA, will be due to global warming. 17 global climate models from CMIP6 under SSP5-8.5 simulate a significant increase in precipitation ($+0.05$ to $0.3 \mp 0.1$ mm day$^{-1}$) over South-Eastern Saharan Desert in North Africa by the end of the century (Arjdal et al., 2023). They also projected that the total soil moisture would increase over Southern Saharan Desert under the SSP5-8.5 (6 to 20%) and SSP2-4.5 (4 to 14%). Based on TWS data from eight global climate models belonging to CMIP6, a broad part of the dry MENA region tends to be wetter under SSP5-8.5 over 2071-2100 (Xiong et al., 2022)."

Line 102: An explanation of "soil moisture z-scores" is needed here.
Reply: Implemented. We added the following explanation to the text.
"It is computed by subtracting the mean and dividing by the standard deviation of the time series from the baseline period".

Line 162: This is covered in the introduction; suggest removing this sentence and moving the second sentence of this paragraph into the preceding paragraphs on the discussion of regional climate.
Reply: Implemented.

Line 174: In addition to defining "real evapotranspiration", please add an explanation on "potential evapotranspiration" and how that is calculated since that is also listed in Table 1. Is real ET a model output and potential ET is calculated from model output?
Reply: Real and potential ET have been calculated. We added the following explanations to the new version:
"Potential evapotranspiration (ET) is the amount of evaporation that would occur if a sufficient water source were available. The Thornthwaite method was used to calculate the potential ET based on the monthly mean temperature and latitude data for each grid. Water evaporation from both soil and canopy and transpiration are summed up to obtain the real ET."

Table 1: The caption mentions historical model output as the data source for this table – please add this to the text as well (e.g., Line 172). It might also make sense for Table 1 to come after the model is introduced in section 2.2.
Reply: We added it to the text as well. However, we believe that Table 1 position in the Study area section is more appropriate.

Line 209: Please clarify the text here – what is meant by "in turn"?
Reply: It means "and consequently". We simply deleted this as follows:
"For the anomaly analysis relative to historical conditions and the multiple linear regression models, we used the first three ensembles of SSP5-8.5 …"

Line 241: Was this correlation calculated or made by eye using the plots in Figure S2? And was the correlation tested for the other variable combinations? Calculating the correlations and reporting them in the paper would make for a stronger justification of the MLR model inputs.
Reply: We have calculated the correlation coefficients. Please see the following results. Except for region 5, the correlation coefficient between TWS and soil moisture in all regions under all scenarios and ensembles are larger than 0.90. The correlation coefficient between temperature and soil moisture in all cases is larger than 0.90 and statistically significant at the 5% level. However, we added these correlation values to the text and also included the following tables in the Supplementary Information.

**Table S2.** The average correlation between the variables under the available ensembles for global warming SSP5-8.5 scenario in the region R1. Consistently, the values inside the parenthesis are the difference-range values between minimum and maximum correlations. The insignificant correlation coefficients (p-value>0.05) are underlined.

| SSP-R1 | TWS | Temp | Precip | SM | RET | PET | LAI |
|---|---|---|---|---|---|---|---|
| TWS | 1 | -0.35 (-0.31, -0.37) | 0.61 (0.58, 0.64) | 0.99 (0.98, 1) | 0.18 (0.15, 0.23) | -0.30 (-0.26, -0.32) | 0.77 (0.72, 0.78) |
| Temp | | 1 | -0.61 (-0.60, -0.62) | -0.35 (-0.32, -0.37) | 0.77 (0.76, 0.78) | 0.96 (0.96, 0.96) | 0.05 (0.02, 0.07) |
| Precip | | | 1 | 0.59 (0.56, 0.63) | -0.17 (-0.13, -0.19) | -0.56 (-0.55, -0.57) | 0.42 (0.40, 0.44) |
| SM | | | | 1 | 0.17 (0.14, 0.22) | -0.30 (-0.26, -0.32) | 0.74 (0.70, 0.77) |
| RET | | | | | 1 | 0.75 (0.75, 0.75) | 0.60 (0.58, 0.63) |
| PET | | | | | | 1 | 0.11 (0.08, 0.13) |
| LAI | | | | | | | 1 |

**Table S3.** As Table S2 but for the region R2.

| SSP-R2 | TWS | Temp | Precip | SM | RET | PET | LAI |
|---|---|---|---|---|---|---|---|
| TWS | 1 | -0.25 (-0.24, -0.26) | 0.50 (0.47, 0.53) | 0.99 (0.99, 1) | 0.54 (0.50, 0.57) | -0.24 (-0.23, -0.25) | 0.72 (0.71, 0.74) |
| Temp | | 1 | -0.48 (-0.47, -0.50) | -0.24 (-0.22, -0.25) | 0.11 (0.10, 0.12) | 0.95 (0.94, 0.96) | -0.18 (-0.15, -0.20) |
| Precip | | | 1 | 0.49 (0.46, 0.51) | 0.53 (0.52, 0.55) | -0.51 (-50, -53) | 0.52 (0.51, 0.53) |
| SM | | | | 1 | 0.53 (0.49, 0.56) | -0.22 (-0.20, -0.24) | 0.71 (0.70, 0.73) |
| RET | | | | | 1 | 0.03 (0.02, 0.04) | 0.85 (0.84, 0.86) |
| PET | | | | | | 1 | 0.16 (0.14, 0.19) |
| LAI | | | | | | | 1 |

**Table S4.** As Table S2 but for the region R3.

| SSP-R3 | TWS | Temp | Precip | SM | RET | PET | LAI |
|---|---|---|---|---|---|---|---|
| TWS | 1 | -0.67 (-0.65, -0.70) | 0.51 (0.49, 0.52) | 0.98 (0.97, 0.99) | 0.34 (0.33, 0.35) | -0.54 (-0.52, -0.57) | 0.80 (0.78, 0.81) |
| Temp | | 1 | -0.83 (-0.82, -0.84) | -0.68 (-0.66, -0.70) | 0.23 (0.22, 0.24) | 0.97 (0.97, 0.97) | -0.44 (-0.42, -0.46) |
| Precip | | | 1 | 0.49 (0.46, 0.51) | -0.28 (-0.27, -0.30) | -0.82 (-0.81, -0.83) | 0.31 (0.29, 0.32) |
| SM | | | | 1 | 0.34 (0.33, 0.35) | -0.55 (-0.53, -0.57) | 0.80 (0.78, 0.81) |
| RET | | | | | 1 | 0.33 (0.32, 0.34) | 0.69 (0.68, 0.70) |
| PET | | | | | | 1 | -0.29 (-0.26, -0.31) |
| LAI | | | | | | | 1 |

**Table S5.** As Table S2 but for the region R4.

| SSP-R4 | TWS | Temp | Precip | SM | RET | PET | LAI |
|---|---|---|---|---|---|---|---|
| TWS | 1 | 0.08 (0.03, 0.11) | 0.19 (0.14, 0.23) | 0.99 (0.98, 1) | 0.35 (0.26, 0.42) | 0.08 (0.05, 0.10) | 0.53 (0.43, 0.59) |
| Temp | | 1 | -0.06 (-0.02, -0.11) | 0.08 (0.04, 0.11) | 0.04 (0.03, 0.06) | 0.92 (0.92, 0.92) | -0.07 (-0.02, -0.12) |
| Precip | | | 1 | 0.18 (0.13, 0.22) | 0.71 (0.68, 0.72) | -0.14 (-0.10, -0.20) | 0.25 (0.23, 0.27) |
| SM | | | | 1 | 0.32 | 0.08 | 0.53 |

| | TWS | Temp | Precip | SM | RET | PET | LAI |
|---|---|---|---|---|---|---|---|
| | | | | | (0.21, 0.40) | (0.06, 0.10) | (0.42, 0.59) |
| RET | | | | | 1 | -0.09 (-0.05, -0.13) | 0.62 (0.60, 0.64) |
| PET | | | | | | 1 | 0.04 (0.01, 0.06) |
| LAI | | | | | | | 1 |

**Table S6.** As Table S2 but for the region R5.

| SSP-R5 | TWS | Temp | Precip | SM | RET | PET | LAI |
|---|---|---|---|---|---|---|---|
| TWS | 1 | 0.15 (0.14, 0.17) | 0.19 (0.15, 0.23) | 0.74 (0.70, 0.76) | 0.29 (0.24, 0.33) | 0.19 (0.18, 0.21) | 0.33 (0.29, 0.35) |
| Temp | | 1 | 0.17 (0.12, 0.24) | 0.31 (0.24, 0.37) | 0.34 (0.31, 0.39) | 0.96 (0.96, 0.96) | 0.50 (0.46, 0.53) |
| Precip | | | 1 | 0.36 (0.31, 0.38) | 0.91 (0.88, 0.93) | 0.12 (0.07, 0.19) | 0.61 (0.54, 0.70) |
| SM | | | | 1 | 0.49 (0.41, 0.55) | 0.32 (0.25, 0.38) | 0.50 (0.43, 0.55) |
| RET | | | | | 1 | 0.30 (0.27, 0.36) | 0.82 (0.79, 0.85) |
| PET | | | | | | 1 | 0.47 (0.44, 0.50) |
| LAI | | | | | | | 1 |

**Table S7.** As Table S2 but for the region R6.

| SSP-R6 | TWS | Temp | Precip | SM | RET | PET | LAI |
|---|---|---|---|---|---|---|---|
| TWS | 1 | 0.29 (0.26, 0.31) | 0.29 (0.24, 0.35) | 0.98 (0.97, 1) | 0.46 (0.38, 0.53) | 0.26 (0.21, 0.31) | 0.26 (0.24, 0.31) |
| Temp | | 1 | 0.06 (0.02, 0.11) | 0.29 (0.26, 0.31) | 0.32 (0.27, 0.39) | 0.95 (0.94, 0.96) | 0.35 (0.28, 0.40) |
| Precip | | | 1 | 0.27 (0.22, 0.32) | 0.84 (0.82, 0.85) | 0.04 (0.02, 0.07) | -0.02 (0.00, -0.06) |
| SM | | | | 1 | 0.43 (0.36, 0.50) | 0.28 (0.26, 0.30) | 0.27 (0.24, 0.32) |
| RET | | | | | 1 | 0.26 (0.23, 0.30) | 0.05 (0.03, 0.11) |
| PET | | | | | | 1 | 0.22 (0.12, 0.29) |
| LAI | | | | | | | 1 |

**Table S8.** As Table S2 but under SAI scenario for the region R1.

| SAI-R1 | TWS | Temp | Precip | SM | RET | PET | LAI |
|---|---|---|---|---|---|---|---|
| TWS | 1 | -0.20 (-0.18, -0.24) | 0.59 (0.58, 0.60) | 0.96 (0.95, 0.97) | 0.13 (0.13, 0.13) | -0.15 (-0.13, -0.18) | 0.52 (0.46, 0.59) |
| Temp | | 1 | -0.51 (-0.49, -0.52) | -0.06 (-0.02, -0.11) | 0.88 (0.87, 0.89) | 0.96 (0.96, 0.96) | 0.44 (0.38, 0.49) |
| Precip | | | 1 | 0.53 (0.52, 0.54) | -0.20 (-0.19, -0.21) | -0.47 (-0.46, -0.49) | 0.26 (0.23, 0.28) |
| SM | | | | 1 | 0.26 (0.25, 0.27) | -0.05 (0.00, -0.08) | 57 (50, 64) |
| RET | | | | | 1 | 0.89 (0.88, 0.90) | 0.76 (0.72, 0.78) |
| PET | | | | | | 1 | 0.52 (0.46, 0.57) |
| LAI | | | | | | | 1 |

**Table S9.** As Table S2 but under SAI scenario for the region R2.

| SAI-R2 | TWS | Temp | Precip | SM | RET | PET | LAI |
|---|---|---|---|---|---|---|---|
| TWS | 1 | -0.26 (-0.24, -0.28) | 0.53 (0.51, 0.55) | 0.97 (0.96, 1) | 0.54 (0.52, 0.57) | -0.25 (-0.23, -0.26) | 0.73 (0.69, 0.79) |
| Temp | | 1 | -0.53 | -0.24 | 0.27 | 0.96 | -0.08 |

| | TWS | Temp | Precip | SM | RET | PET | LAI |
|---|---|---|---|---|---|---|---|
| | | | (0.51, 0.55) | (-0.22, -0.26) | (0.26, 0.28) | (0.96, 0.96) | (-0.05, -0.09) |
| Precip | | | 1 | 0.51 (0.50, 0.53) | 0.41 (0.40, 0.42) | -0.57 (-0.54, -0.59) | 0.50 (0.48, 0.51) |
| SM | | | | 1 | 0.53 (0.51, 0.56) | -0.23 (-0.21, -0.24) | 0.72 (0.68, 0.78) |
| RET | | | | | 1 | 0.21 (0.21, 0.22) | 0.84 (0.84, 0.84) |
| PET | | | | | | 1 | -0.04 (-0.03, -0.05) |
| LAI | | | | | | | 1 |

**Table S10.** As Table S2 but under SAI scenario for the region R3.

| SAI-R3 | TWS | Temp | Precip | SM | RET | PET | LAI |
|---|---|---|---|---|---|---|---|
| TWS | 1 | -0.54 (-0.53, -0.55) | 0.43 (0.42, 0.44) | 0.98 (0.97, 0.99) | 0.25 (0.24, 0.26) | -0.46 (-0.46, -0.47) | 0.76 (0.75, 0.77) |
| Temp | | 1 | -0.78 (0.78, 0.79) | -0.54 (-0.53, -0.55) | 0.55 (0.54, 0.56) | 0.98 (0.96, 0.99) | -0.18 (-0.16, -0.19) |
| Precip | | | 1 | 0.42 (0.41, 0.43) | -0.37 (-0.35, -0.38) | -0.78 (-0.78, -0.78) | 0.18 (0.17, 0.21) |
| SM | | | | 1 | 0.26 (0.25, 0.26) | -0.46 (-0.46, -0.47) | 0.76 (0.75, 0.78) |
| RET | | | | | 1 | 0.59 (0.58, 0.59) | 0.67 (0.66, 0.67) |
| PET | | | | | | 1 | -0.09 (-0.07, -0.10) |
| LAI | | | | | | | 1 |

**Table S11.** As Table S2 but under SAI scenario for the region R4.

| SAI-R4 | TWS | Temp | Precip | SM | RET | PET | LAI |
|---|---|---|---|---|---|---|---|
| TWS | 1 | 0.03 (0.01, -0.04) | 0.28 (0.27, 0.30) | 0.99 (0.98, 1) | 0.45 (0.37, 0.52) | 0.03 (0.02, 0.04) | 0.56 (0.52, 0.60) |
| Temp | | 1 | -0.05 (-0.02, -0.07) | 0.03 (0.01, 0.04) | 0.09 (0.06, 0.10) | 0.94 (0.94, 0.94) | 0.16 (0.10, 0.24) |
| Precip | | | 1 | 0.27 (0.26, 0.29) | 0.78 (0.76, 0.79) | -0.14 (-0.11, -0.15) | 0.32 (0.26, 0.36) |
| SM | | | | 1 | 0.43 (0.36, 0.50) | 0.03 (0.02, 0.04) | 0.55 (0.51, 0.59) |
| RET | | | | | 1 | -0.02 (0.00, -0.03) | 0.68 (0.67, 0.69) |
| PET | | | | | | 1 | 0.15 (0.08, 0.23) |
| LAI | | | | | | | 1 |

**Table S12.** As Table S2 but under SAI scenario for the region R5.

| SAI-R5 | TWS | Temp | Precip | SM | RET | PET | LAI |
|---|---|---|---|---|---|---|---|
| TWS | 1 | 0.29 (0.22, 0.34) | 0.15 (0.14, 0.16) | 0.73 (0.64, 0.80) | 0.28 (0.26, 0.31) | 0.33 (0.27, 0.38) | 0.36 (0.28, 0.41) |
| Temp | | 1 | 0.12 (0.12, 0.13) | 0.29 (0.24, 0.34) | 0.36 (0.35, 0.37) | 0.97 (0.97, 0.97) | 0.56 (0.52, 0.59) |
| Precip | | | 1 | 0.37 (0.33, 0.41) | 0.88 (0.88, 0.89) | 0.09 (0.09, 0.10) | 0.56 (0.52, 0.58) |
| SM | | | | 1 | 0.54 (0.50, 0.58) | 0.32 (0.26, 0.37) | 0.50 (0.46, 0.54) |
| RET | | | | | 1 | 0.34 (0.32, 0.35) | 0.80 (0.77, 0.81) |
| PET | | | | | | 1 | 0.56 (0.52, 0.59) |
| LAI | | | | | | | 1 |

**Table S13.** As Table S2 but under SAI scenario for the region R6.

| SAI-R6 | TWS | Temp | Precip | SM | RET | PET | LAI |
|---|---|---|---|---|---|---|---|
| TWS | 1 | 0.11 (0.06, 0.17) | 0.26 (0.19, 0.31) | 0.99 (0.98, 1) | 0.44 (0.38, 0.50) | 0.11 (0.05, 0.16) | 0.33 (0.31, 0.35) |
| Temp | | 1 | 0.06 (0.05, 0.08) | 0.11 (0.06, 0.17) | 0.35 (0.33, 0.37) | 0.96 (0.96, 0.96) | 0.37 (0.34, 0.40) |
| Precip | | | 1 | 0.24 (0.18, 0.29) | 0.73 (0.72, 0.74) | 0.00 (0.00, 0.00) | 0.08 (0.03, 0.12) |
| SM | | | | 1 | 0.42 (0.36, 0.47) | 0.10 (0.05, 0.16) | 0.33 (0.31, 0.34) |
| RET | | | | | 1 | 0.29 (0.26, 0.31) | 0.26 (0.19, 0.30) |
| PET | | | | | | 1 | 0.32 (0.30, 0.35) |
| LAI | | | | | | | 1 |

Line 246: How often / how many outliers were removed using this method?
Reply: We totally fitted 36 models. The number of outliers removed is specific for each model. In some models, no outliers were detected while in others up to 5 data points were removed. We added the following sentence to the manuscript:
"The number of outlier data points excluded varies from zero to 5 in the 36 models."

Line 254: Please add more details on the method here – specifically what is meant by "independent variable-order average over average contributions…" and "impacts adjusted for other regressors".
Reply: "independent variable-order average over average contributions…" refers to the fact that the order in which variables are added to the model can affect how much each variable contributes to the overall R-squared. If a variable that has a strong linear relationship with the dependent variable be initially added, it may capture a significant portion of the variance, which can affect the subsequent contributions of other variables. To mitigate the order dependency, the LMG calculates average contributions by considering all possible orders and then averaging the results. This approach helps provide a more robust estimate of variable importance that is less sensitive to the specific order in which variables are considered. For clarity we revised this sentence as follows:
"The LMG method considers the average contributions of each variable across different model sizes and then averages these averages to provide a more robust measure of variable importance."

To clarify the "impacts adjusted for other regressors" term we have added the following additional description to the text:
"As the considered variables may be correlated with each other, when a new predictor is added to a model that already contains other predictors, its impact can be influenced by the presence of those other variables. The LMG method takes into account these interactions and adjusts the variable's contribution to reflect its unique impact while considering the effects of other regressors."

Figure 2: It is difficult to see the trends in the anomalies with the strong seasonal cycle in certain regions. Suggest removing the seasonal cycle and/or some other method of filtering out noise here (e.g., running yearly means).

Reply: We have also extracted the long-term trends using Singular Spectrum Analysis (SSA) method (below figure), consistent with the results from Figure 2. This figure has also been included in the Supplementary Information.

[Figure]

**Figure RC1-1.** The long-term trends of TWS anomaly relative to the TWS averaged over the historical period across MENA and the lands around the Caspian and Mediterranean Seas under global warming without (SSP5-8.5) and with SAI (SSP5-8.5-SAI). Figures a-f respectively are for regions R1 to R6. Shading in each curve shows the across-ensemble range. The dashed line crossing the *y*-axis at zero in each subplot is the ensemble mean of TWS over the historical period (1985-2014).

Lines 282-295: Please use the region labels (e.g., R1-R6) in the text here to ease the interpretation of Figure 3.
Reply: Implemented (please see new Figure 3 on the next page).

Line 292: Clarify "Mean TWS" here – is that the temporal mean, ensemble mean, spatial mean, or some combination?
Reply: It refers to temporal-ensemble mean.

Figure 3 (and others): For the difference plots (e.g., panels b-d), I recommend choosing a different color scale with a clear divergence at 0. With this yellow/blue color scale it is difficult to discern positive vs. negative regions of change. Same comment for Figures S2, S4. The color scale in Figure S1 works better for difference plots.
Reply: Implemented. As an example, please see new Figure 3.

[Figure]

**Figure 3.** Ensemble mean maps of TWS across the studied domain in the historical climate (a) over 1985-2014 and their projected future changes in the 2071–2100 period under the SSP5-85 GHG scenario (SSP5-8.5 minus historical (b) and GHG+SAI minus historical (c)). The extent to which the SAI impacts the TWS changes imposed by global warming is further shown (SAI minus SSP5-8.5 (d)). Hatched areas show where all ensemble members agree on the sign of the changes.

Lines 304-311: Use percentage values instead of absolute kg/m^2 changes in the text here to match the black labels in Figure 4. Or use absolute labels in Figure 4, which would match the y-axis.

Reply: We have used percentage in the text in the new version as follows:

"Mean TWS significantly ($p<0.05$) decreases in the wetter lands around the Caspian (R1) and Mediterranean (R3) Seas to the north (3.7-5.2% on area average) while it significantly increases in the dry region of Arabian Peninsula (5.6%) in response to GHG warming."

Figure 4: In addition to the partial reversals (R1, R3, R4) and the overcompensation (R2), SAI also has an amplifying effect in R5 and a slight overcompensation in R6 – it is worth noting these responses in the text (even if to say they are not significant).

Reply: The following sentence has been added to the text:

"SAI also has an amplifying effect in R5 and a slight overcompensation in R6, but its impact is statistically insignificant."

Figure 4: Why are there three p-values shown at the bottom of each panel? I assume two of the values denote the significance of the changes in SSP and SAI relative to historical, but what does the other value represent? Please clarify in the figure caption. Same comment for Figures S3 and S5.

Reply: Implemented. The new caption for Figure 4 is copied below:

"Figure 4. Box and whiskers plot of the changes in the Terrestrial Water Storage (TWS) in regions 1 to 6 over 2071-2100 under SSP5-8.5 and SSP5-8.5-SAI relative to historical conditions (1985-2014). The titles of each subplot refer to the regions. The median for each experiment is denoted by the red line, the upper (75th) and lower (25th) quartiles by the top and bottom of the box, and ensemble limits by the whisker extents. The positive/negative values in black are the change percent under SSP5-8.5 and SSP5-8.5-SAI relative to the median of the historical period data. The three values in green refer to p-values between historical and global warming, historical and SAI, and global warming and SAI, respectively, obtained from t-test analysis in which the underlined p-values are statistically significant."

Line 330: Similar question to Line 241 – was significance calculated here or by eye? Why do non-overlapping curves imply significance?

Reply: We first conducted the repeated measures analysis of variance which compares means across one or more variables that are based on repeated observations, and then performed post hoc Tukey-Kramer comparisons to determine which curves (including its upper and lower bounds) are significantly different from each other (please see the new Figure 5 below). However, we have added the above explanations to the text.

[Figure]

**Figure 5.** The TWS anomaly return level versus return period using the first three realizations for the historical, SSP5-8.5, and SSP5-8.5-SAI in regions 1 to 6 (a to f). The two parallel dashed black lines refer to 30- (left) and 50-year (right) return periods. Shading in each curve is the 95% upper and lower confidence bands. The three values in red refer to *p*-values between historical and global warming, historical and SAI, and global warming and SAI, respectively, obtained from the repeated measures analysis of variance and the post hoc Tukey-Kramer comparisons in which the underlined *p*-values are statistically significant.

Lines 344-360: Please use the region labels R1-R6 in text here to ease comparison to Table 2.

Reply: Implemented. Please see the revised version of this part copied below:

"Global warming, on the whole, decreases the TWS extremes (i.e., fewer wetter conditions) at 30- to 100-year return periods over all the study areas except for the Arabian Peninsula (R4) and western NA (R6). The most robust decreases in the extreme TWS imposed by global warming relative to historical conditions occur in the lands around the Caspian R1 (-108% on average over return periods from 30- to 100-year) and Mediterranean R3 (-43% on average) and the eastern NA R5 (-89% on average) are partially suppressed by SAI. A small increase in the extreme TWS in Iran and Iraq (R2) simulated under GHG (+15%) is overcompensated by SAI (+57%)."

Line 346 and following: Please clarify "decreases the TWS extremes" – does this mean a decrease in positive extremes (i.e., fewer wetter conditions) or negative extremes (fewer drier conditions) or both?

Reply: Note that return levels refer to the peaks in the TWS time series. Since here we used the anomalies relative to the historical mean, the peaks may be negative or positive. However, here we mean the fewer wetter conditions. Please see the following revised part:

"Global warming, on the whole, decreases the TWS extremes (i.e., fewer wetter conditions) at 30- to 100-year return periods …".

Lines 378-380: Please clarify here whether this is referring to the most important variable under SAI or SSP (or both).

Reply: Under both SSP and SAI.

Lines 386-389: Please clarify what is meant by "due to evapotranspiration" if this is looking only at temperature and precipitation ("with just temperature and precipitation as independent variables"). Are there results that look at subsets of these three variables and are they included somewhere?

Reply: Our mean of "With just temperature and precipitation as independent variables, …" is that if we compare the temperature importance role on TWS with precipitation role. We just want to know between precipitation and temperature which one is more important. However, in all TWS MLR models, all four variables of real ET, precipitation, temperature, and leaf area index (i.e., vegetation coverage) have been considered.

Line 399: Please include the specific variance explained values for the MLR models somewhere in the text or figures (e.g., the bars of Figure 6-7).

Reply: We have included the ensemble-mean variance on Figures 6-7.

[Figure]

**Figure 6.** LMG importance plot (Lindeman et al., 1980) of the four independent variables in the regression for TWS for the global warming SSP5-8.5 scenario in each region. The bar and range-bar respectively show the ensemble mean importance and the range of importance from the three ensemble members. The three values in red on each subplot shows the minimum, mean, and maximum variances explained by models.

[Figure]

**Figure 7.** As in Fig. 6, but for the SSP5-8.5-SAI scenario.

Figures 6-7: Here, or perhaps in the methods section, please provide some context for the importance values (*y*-axis). Is this unitless, and if so, should the individual variable contributions total to 1 if all the appropriate variables were sampled? Are interactions considered?

Reply: Implemented (please see new Figures 6 and 7 above). Importance is a unitless variable and the sum of all independent variable importance's in each model equals the model's variance explained. The importance values for individual variables do not necessarily need to total to 1. However, we have added the following sentence to the text:

"Importance is a unitless variable and the sum of all independent variable importance's in each model equals the model's explained variance."

In the case of LMG, it's a method that explicitly considers interaction effects and decomposes the total variance explained into contributions from individual variables, pairs of variables, and higher-order interactions. Therefore, LMG importance values can provide insights into both main effects and interaction effects.

Lines 444-457: Most of this paragraph should go in the results section, as the supplemental figures have not yet been discussed. The last sentence gets to a comparison with other studies which is appropriate for the discussion section and can be merged with another paragraph.

Reply: We have moved this part into the results, last part of section 3.1, as copied below:

"We also compared the changes in TWS with changes in precipitation, temperature, real ET, soil moisture, and potential ET over each region under both global warming and SAI scenarios (Figs. S2 to S6 in the Supplementary Information). The TWS decreasing patterns under both SSP5-8.5 and SSP5-8.5-SAI scenarios across the entire study area are similar to soil moisture change patterns (Fig. S2 and S4 in Supplementary Information) but more widespread than precipitation under global warming (Fig. S4). The decreased TWS is seen beyond the regions of reduced precipitation (Fig. S4), from beyond the Mediterranean and Atlantic coasts to include Syria, Iraq, and the lands around the Caspian Sea as well as to a wide portion of NA (Fig. 4). These include places where precipitation is either increasing or shows no significant change, consistent with results reported by Cook et al. (2020).

In Summary, our findings show that the SSP5-8.5-SAI scenario has a potential to partially offset the significant changes in mean TWS imposed by SSP5-8.5 over the entire MENA. While SAI (Fig. 3d) succeeded in reversing mean TWS deficits in the wetter lands around the Caspian and Mediterranean Seas driven by the GHG SSP5-8.5 scenario (Fig. 3b), it did not fully cancel out the TWS deficits (Figs., 3c, 4a, and 4c). However, in the dry MENA regions (Fig. 3d), particularly Iran (containing the Lut desert in the central region and the Kavir desert in the east), Iraq, and the Arabian Peninsula (housing the Arabian Desert), SAI resulted in higher mean water storage relative to the historical period (Figs. 3c and 4)."

Line 446: Please specify which simulation "The TWS decreasing patterns" refers to.

Reply: Both SSP5-8.5 and SSP5-8.5-SAI scenarios.

Line 461: Related to vegetation, it is worth discussing the competing impacts of high CO2 and less solar radiation in the SAI scenario. These impacts could also be contributing to the overall ET, soil moisture, and TWS responses. The regions discussed here have varying amounts of vegetation and that could be contributing to the range of regional responses.

Reply: It has been found that considering vegetation variable for TWS leads to improved TWS model output (Trautmann et al., 2022). Plants absorb water from the soil and release it into the atmosphere through transpiration. Hence, we added leaf area index (LAI) as a new variable into MLR models and results (please see new Figures 6 and 7 above and Figures S6 and S7 below). We also used LAI findings in the discussion as copied below:

"MENA houses several deserts such as Saharan in NA, Arabian in the Arabian Peninsula, and Lut and Kavir in Iran. As a result, it features several dust hotspots (Mousavi et al., 2023). One noteworthy aspect of our findings is the projection of higher water storage, increased soil moisture, and enhanced vegetation coverage under the solar geoengineering (SAI) scenario relative to historical conditions. This evidence holds the promise of potentially reducing dust concentrations in MENA in the future, in line with reduced dust concentrations projections made by Mousavi et al. (2023) using the GLENS project.

The more robust and widespread deficit in mean TWS compared to precipitation in the area, which is in line with results reported by Cook et al. (2020), highlights the profound roles that other variables/processes have on the increased ET such as greater atmospheric moisture demand (Dai et al., 2013, 2018) and greater vegetation water use (Mankin et al., 2019) owing to warmer conditions under global warming, consistent with regression model results. According to MLR model results (Figs. 6 and 7), the projected changes in TWS were not solely attributable to precipitation; its interplay with other factors, such as vegetation coverage, temperature, and evapotranspiration play a pivotal role. The vegetation coverage as the primary variable influencing changes in TWS within the area (except for the eastern NA) substantially increases under global warming (Figures RC2 and RC3). It has an important, but often complex and uncertain, role in surface water content (Lemordant et al., 2018; Trugman et al., 2018); the denser vegetation coverage, the higher evapotranspiration rates."

[Figure]

**Figure S6**. As Figure 3 but for LAI.

[Figure]

**Figure S7**. As Fig. 4 but for LAI.

Figure S3: I thought the middle row of this plot (TWS) would be same as Figure 4, but it appears to be different. What is plotted here and what is the difference with Figure 4?
Reply: Agreed. We rechecked and replotted Figures S3 and 4, now they are the same. It seems that in the previous version of Figure S3 was an initial plot after which I edited some parts of the code.

[Figure]

**Figure S3.** Box and whiskers plot of the changes in soil moisture (upper row), terrestrial water storage (TWS, middle row), and potential evapotranspiration (ET, bottom row) in each region from R1 to R6 row). The titles of each subplot refer to the regions. The median for each experiment is denoted by the red line, the upper (75th) and lower (25th) quartiles by the top and bottom of the box and ensemble limits by the whisker extents. The positive/negative values in black are the change percent relative to the median of the historical 20th period data for each variable, respectively. The three values refer to p-values between historical and global warming, historical and SAI, and global warming and SAI, respectively, obtained from t-test analysis in which the underlined p-values are statistically significant.

Figure S4: For the middle row (temperature) difference plots, the color bar limits should be increased on both ends to better show the regional responses.

Reply: Various limits have been tested; the following is the best.

[Figure]

**Figure S4**. As in Figure S2, but for the variables of precipitation (upper row), surface temperature (middle row), and real evapotranspiration (ET, bottom row).

Data availability: Suggest providing some more information on how to access these specific CESM simulations via the ESGF website (e.g., Source ID, Experiment ID). Tilmes et al. 2020 also has a DOI for the SAI simulations which should be included if those experiments are not on ESGF: https://doi.org/10.26024/t49k-1016.

Reply: we rewrite the data availability as follows:
"The data for CESM2 simulations are publicly available via its website: https://esgf-node.llnl.gov/search/cmip6/. To access these specific data via ESGF website use the Source ID = CESM2-WACCM, Experiment ID=ssp585, and Frequency = mon. The SSP5-8.5-SAI data are freely available at https://www.earthsystemgrid.org/dataset/ucar.cgd.ccsm4.geomip.ssp5.html (https://doi.org/10.26024/t49k-1016)."

Technical corrections:
Line 39: Typo "Projected" should not be capitalized.
Reply: Implemented.

Lines 335-336: I think this should be "return levels" instead of "level returns".
Reply: Implemented.

Lines 335-337: Should these sentences be combined?
Reply: Implemented.

Figure 5: Please add panel labels to the subplots and update caption to "(a to f)".
Reply: Done (please see new Figure 5 above).

Line 397: Typo "EV"
Reply: Corrected.

Line 467: Typo "EV"
Reply: Corrected.

Lines 520-522: I think "SAI" is missing after "with...and without" here.
Reply: Added.

---

## Author Comment (AC2)

**Response to RC2:**

Thank you for inviting me for reading this article. The authors evaluate terrestrial water storage under SSP5-8.5 and SSP5-8.5-SAI scenarios across the Middle East and North Africa. The results are useful for supporting aerosol intervention strategy against global warming and water resources management for Mediterranean, Middle East, and North Africa. I have some concerns about the methods and figures which may be helpful for improvement.

Reply: We sincerely appreciate your effort and time in reviewing our manuscript as well as your constructive comments/suggestions. We have made every effort to incorporate your feedback effectively. Below, you will find a detailed response to each comment, with comments presented in black and our responses in red.

1- Section 2.3. The authors calculate return periods from GEV distribution. However, GEV distribution is used to simulate maximum value in a certain period, instead of monthly values. The authors may give more details about how to apply GEV distribution. Did the authors calculate the annual maximum TWS values? In addition, authors may provide empirical probabilities and examine whether annual maximum TWS follows GEV distribution or other distributions.

Reply: We applied a GEV distribution to the complete dataset of monthly TWS values without explicitly setting maximum values. This approach allowed us to estimate the parameters of the GEV distribution using the entire dataset. However, in response to your request, we have also extracted the annual maximum TWS values and provided the corresponding fitted GEV distribution for comparison with the full dataset scenario (e.g., Figures RC2-1 and RC2-2 below).

Overall, the probability densities for both datasets exhibit a high degree of similarity across various regions and scenarios. For instance, Figures RC2-1 and RC2-2 illustrate the probability densities for the R2 and R5 regions. Additionally, the graphs depicting return levels versus return periods based on annual maximums (Figure RC2-3) closely resemble the results obtained from the entire dataset (Figure 5). In all cases, the trends are highly similar (compare Figure 5 and Figure RC-3), although it's worth noting that the annual maximums scenario exhibits slightly wider upper and lower bounds compared to the entire dataset scenario. Regarding the significance test for differences between historical, global warming, and SAI scenarios, the results are consistent across all cases. However, there is an exception in the case of the difference between historical and SAI scenarios in R5. In the entire dataset scenario, a significant difference is observed (Fig. 5e), whereas in the annual maximums case, it does not reach significance (Fig. S9e).

In light of these explanations, we have retained the results obtained from the entire dataset in the main text, and we will include the results from the annual maximums scenario in the Supplementary Materials.

[Figure]

**Figure RC2-1.** Probability density curves for Region R2, comparing two scenarios: one using all available data (left column) and the other using annual maximum values (right column) under the historical conditions (upper row) as well as the GHG emissions (middle row) and SAI (lower row) scenarios for region R2.

[Figure]

**Figure RC2-1.** As Figure RC2-1 but for the region R5.

[Figure]

**Figure RC2-3.** As in Figure 5 but for the annual maximums.

2- Line 211. The historical period is from 1985-2014, and future period is from 2071-2100. The authors do not analyze mid-21th century. The authors may explain why you do not analyze the full period from 1850-2100.

Reply: Agreed. We have included the following clarifications in the text:

"We have chosen to focus on the 2071-2100 future period because the anticipated changes in TWS driven by GHG emissions are expected to be more pronounced during this time frame (Pokhrel et al., 2021). Our decision to prioritize this period is based on the need to examine the effects of more significant alterations resulting from global warming."

"We focused on the historical period from 1985 to 2014 rather than the entire historical dataset spanning from 1850 to 2100 for several reasons. Firstly, recent historical climate data may exhibit less uncertainty, given that additional meteorological stations with improved data quality are available to be used for model calibrations (Zhang et al., 2020). Secondly, this selected historical period offers valuable insights into the observable impacts of climate change, which are highly pertinent to present-day societal and environmental challenges. These insights are of utmost importance to policymakers and communities alike. Thirdly, the chosen historical 30-year time period aligns with the 30-year periods considered for the GHG emissions and SAI scenarios, ensuring consistency in our statistical analysis."

Ref:

Pokhrel, Y., Felfelani, F., Satoh, Y., Boulange, J., Burek, P., Gädeke, A., ... & Wada, Y. (2021). Global terrestrial water storage and drought severity under climate change. Nature Climate Change, 11(3), 226-233.

Zhang, B., Xia, Y., Long, B., Hobbins, M., Zhao, X., Hain, C., ... & Anderson, M. C. (2020). Evaluation and comparison of multiple evapotranspiration data models over the contiguous United States: Implications for the next phase of NLDAS (NLDAS-Testbed) development. Agricultural and Forest Meteorology, 280, 107810.

3- Authors only select CESM2 for analysis. The authors may evaluate the performance CESM2 for historical climate over the study area to validate this model.

Reply: We have incorporated the following information into the new version:

"For global terrestrial ET, the CESM2(WACCM) ranked as the second-best model among 19 CMIP6 models (Wang et al., 2021)."

"In the evaluation by Babaousmail et al. (2021), which assessed 15 CMIP6 models in replicating monthly rainfall patterns spanning from 1951 to 2014 in NA, CESM2(WACCM) emerged as one of the top-performing models. It accurately captured rainfall peaks across the region, albeit with a slight overestimation (ranging from 5 to 10 mm/month) in the southern areas and a slight underestimation (ranging from 0 to 20 mm/month) in the northern regions. Despite these minor deviations, CESM2(WACCM) was recognized as one of the better models for simulating precipitation patterns across North America, achieving a Taylor skill score (TSS) of 0.62. Evaluation of CESM2(WACCM) across the Mediterranean coasts placed it at the 9th and 17th positions out of 31 CMIP6 models for its performance in simulating temperature and precipitation (Bağçaci et al. (2021). Furthermore, when it comes to simulating precipitation relative to observational data for northeastern Iran during the period of 1987-2005, CESM2 stood out as the top-performing model among six CMIP6 models (Zamani et al., 2020). Assessing the representation of spatial and temporal variations in historical

precipitation from 1980 to 2014 across Africa and the Arabian Peninsula, the CMIP6 multi-mean ensemble (inclusive of CESM2-WACCM) demonstrated reasonable performance, as highlighted in Nooni et al. (2023)."

Ref:
Babaousmail, H., Hou, R., Ayugi, B., Ojara, M., Ngoma, H., Karim, R., ... & Ongoma, V. (2021). Evaluation of the performance of CMIP6 models in reproducing rainfall patterns over North Africa. Atmosphere, 12(4), 475.

Bağçaci, S. Ç., Yucel, I., Duzenli, E., & Yilmaz, M. T. (2021). Intercomparison of the expected change in the temperature and the precipitation retrieved from CMIP6 and CMIP5 climate projections: A Mediterranean hot spot case, Turkey. Atmospheric Research, 256, 105576.

Nooni, I.K.; Ogou, F.K.; Chaibou, A.A.S.; Nakoty, F.M.; Gnitou, G.T.; Lu, J. (2023). Evaluating CMIP6 Historical Mean Precipitation over Africa and the Arabian Peninsula against Satellite-Based Observation. *Atmosphere*, *14*, 607. https://doi.org/10.3390/atmos14030607

Wang, Z., Zhan, C., Ning, L., & Guo, H. (2021). Evaluation of global terrestrial evapotranspiration in CMIP6 models. Theoretical and Applied Climatology, 143, 521-531.

Zamani, Y., Hashemi Monfared, S. A., Azhdari Moghaddam, M., & Hamidianpour, M. (2020). A comparison of CMIP6 and CMIP5 projections for precipitation to observational data: the case of Northeastern Iran. Theoretical and Applied Climatology, 142, 1613-1623.

4- Authors use the MLR model to predict TWS. Apart from potential ET, the actual ET is also correlated with temperature and precipitation. How to solve the collinearity between ET, temperature and precipitation?

Reply: In assessing collinearity, we employed the VARCLUS procedure, a method that partitions a set of numeric variables into distinct or hierarchical clusters (Sarle, 1990). Each cluster is associated with a linear combination of the variables it contains. The criterion in this procedure is that if the proportion of the variance explained by a cluster is larger than 0.8 (Figures RC2-4 and RC2-5 below), we should choose one variable from that cluster. It's worth noting that there was minimal variation among ensemble members for each scenario across regions. As a result, we have exclusively presented results for the ensemble r1 in Figures RC2-4 and RC2-5. Based on our findings (refer to Figures RC2-4 and RC2-5 below), in most instances, we needed to select one variable from pairs like potential ET and temperature or TWS and soil moisture. Consequently, we opted for temperature and TWS for our analysis. However, in the arid regions R4 to R6, although both real ET and precipitation were categorized within a single cluster, the proportion of variance explained by the cluster fell below 0.8. Hence, we decided to consider both variables. In response to the comment made in RC1, we also included leaf area index (LAI) as an additional variable in our analysis.

Incorporating above additional explanations into Section 2.4 of the methodology, it would read as follows:

"We employed the VARCLUS procedure to thoroughly assess collinearity among the variables. VARCLUS is a method that effectively segregates a set of numeric variables into disjoint or hierarchical clusters, each characterized by a linear combination of the variables within the cluster (Sarle, 1990). The criterion is that when the proportion of the variance explained by a cluster is larger than 0.8, it is advisable to select one variable from that cluster.

Based on the results obtained from VARCLUS, we made specific decisions to enhance the robustness of our analysis. For instance, we identified strong correlations exceeding 0.9 between potential ET and temperature, as well as between soil moisture and TWS. Consequently, we chose to exclude potential ET and soil moisture from our analysis due to their high levels of correlation with temperature and TWS, respectively."

Ref:
Sarle, W. (1990). The VARCLUS Procedure. In *SAS/STAT User's Guide* (fourth, Vol. 2, pp. 1641–1659). SAS Institute, Inc.
http://support.sas.com/documentation/onlinedoc/stathttp://support.sas.com/documentation/onlinedoc/stat

[Figure]

[Figure]

**Figure RC2-4.** This tree diagrams illustrate the cluster hierarchy within ensemble r1 of the SSP5-8.5 scenario across regions R1 to R6. The y-axis represents the Proportion of Variance Explained.

[Figure]

[Figure]

**Figure RC2-5.** As in Figure RC2-4 but for the SSP5-8.5-SAI scenario.

5- Authors remove outliers in the MLR model. This will artificially give better results. Please justify the removal of these values?

Reply: Overall, the maximum number of outliers removed (5) is relatively insignificant when considering the total number of records in each timeseries (which exceeds 700 in our study). Therefore, it is unlikely to have a substantial impact on the model. Nonetheless, we have incorporated the following statement into the text:

"The number of outliers data points excluded varies from zero to 5 (of the 700 points) in the 36 models"

6- The temporal autocorrelation is an important component in TWS evolution. Monthly TWS is not only impacted by concomitant precipitation and temperature, but also antecedent soil moisture and climatic variables. Authors may consider include climatic variable in previous months as predictors as well.

Reply: In our models, we excluded soil moisture from the list of predictor variables due to its collinearity with TWS. Additionally, we conducted a temporal autocorrelation analysis on all the variables, including temperature, precipitation, real ET, and LAI data for each model. This analysis was carried out using the Autocorrelation function at a 95% confidence level.

In all regions (except R4), the autocorrelation results indicated that the lags at the first and second months were statistically significant, while the third month lag was almost non-significant. Therefore, we modified the LMS model to include information from the two preceding months in these regions.

However, in region R4, we observed different patterns. In this region, both real ET and temperature significantly depended on their respective conditions from the two previous months, while precipitation did not show this effect. Moreover, TLAI in R4 exhibited dependencies on the first three and four preceding months under the SSP and SAI scenarios, respectively. Consequently, we incorporated specific lagged months for each variable in R4.

We have included the updated figures (Figures 6 and 7) below to reflect these changes. Furthermore, we will revise the MLR methodology and Section 3.3 in accordance with this information.

[Figure]

**Figure RC2-6**. The autocorrelation plot for real ET in region R4 under the SAI scenario, specifically ensemble member 003. The y-axis represents lag values in terms of months.

[Figure]

**Figure RC2-7.** As in Figure RC2-6 but for TLAI.

[Figure]

**Figure 6.** LMG importance plot (Lindeman et al., 1980) of the four independent variables in the regression for TWS for the global warming SSP5-8.5 scenario in each region. The bar and range-bar respectively show the ensemble mean importance and the range of importance from the three ensemble members. The three values in red on each subplot shows the minimum, mean, and maximum variances explained by models.

[Figure]

**Figure 7.** As in Fig. 6, but for the SSP5-8.5-SAI scenario.

7- Water storage include soil moisture, groundwater, snow, ice, and others. Figure S3 seems to indicate soil moisture is the dominant driver of TWS variations. It may be insightful for evaluate the relative contributions of other components in TWS.

Reply: In the CMIP6 climate models, TWS is defined as the sum of snow water equivalent and soil moisture (Wu et al., 2021). Consequently, it is reasonable to assume that soil moisture plays a dominant role in driving TWS variations, especially in arid regions. We have incorporated this information into the text with the following sentence:

"TWS is the sum of snow water equivalent and soil moisture (Wu et al., 2021). In the drier regions the soil moisture variability accounts for the dominant component of TWS variability (Pokhrel et al., 2021)"

Ref:
Wu, R. J., Lo, M. H., & Scanlon, B. R. (2021). The annual cycle of terrestrial water storage anomalies in CMIP6 models evaluated against GRACE data. Journal of Climate, 34(20), 8205-8217.

8- Figure S4 is important for interpreting current results. May consider to place this figure in main text.

Reply: Understood. Since the primary focus of the study is on Terrestrial Water Storage (TWS), it will be kept in the supplementary materials.

9- It may be useful to compare the results with previous evaluations (https://www.nature.com/articles/s41558-020-00972-w; Global terrestrial water storage and drought severity under climate change).

Reply: Implemented. We used it in our discussions as follows:

"The CMIP5 outputs also confirm that the global warming (RCP2.6 and RCP6.0) substantially decreases the TWS in the Mediterranean by the mid- (2030-2059) and late- (2070-2099) twenty-first century (Pokhrel et al, 2021)."

"Similarly, a decrease in precipitation (Kim and Byun, 2009), surface runoff (Cook et al., 2020), and TWS (Pokhrel et al., 2021) has been reported across Mediterranean coasts under GHG warming."

Ref:
Pokhrel, Y., Felfelani, F., Satoh, Y., Boulange, J., Burek, P., Gädeke, A., ... & Wada, Y. (2021). Global terrestrial water storage and drought severity under climate change. Nature Climate Change, 11(3), 226-233.

10- Line 29, this sentence may be improved. May explain "more continental" and "hyper-arid" climates? Specify what is different response?

Reply: Implemented. We edited the text as follows:

"… with hyper-arid climate (with annual precipitation less than 100 mm) has the lowest precipitation, real ET, soil moisture, and TWS. More continental areas have characteristics that are typical of continental climates and are less influenced by the moderating effects of nearby oceans."

To clarify the distinctive response observed in R5, we included the following information in the text:

"Unlike the other arid regions, in eastern NA (R5), we observe a reduction in the mean TWS trend under both GHG and SAI scenarios, and the extreme TWS values are also lower compared to the historical conditions."

11- Line 86, may place this paragraph earlier than the introduction of SRM, which is proposed to address climate change.

Reply: Implemented.

12- Line 127. What is the regional consequence and hydrological cycle? May give more explanations

Reply: We revised it as follows:

"While SAI may counteract the annual-mean water availability changes over land forced by GHG, it is not easy to offset the regional consequences, especially in the hydrological cycle, such as the Amazonian drying trend and the reduced precipitation (P), evaporation (E), and P-E (Jones et al., 2018)."

13- Line 228 and Eq. (1). The authors give the equations for Xi = 0 in equation (2). It may be better to provide CDF when Xi ($\xi$)= 0 in Equation (1) as well. In addition, I think Eq.(1) is the CDF instead of PDF. It is better to clearly specify this.

Reply: Implemented. We added mor explanations to this part as follows:

"The GEV probability density and cumulative distribution functions are defined as (Gilleland, 2020):

$$g(z) = \frac{1}{\sigma} t(z)^{1+\xi} e^{-t(z)}; \quad G(z) = e^{-t(z)}; \quad t(z) = \begin{cases} \left\{ 1 + \xi \left( \frac{z-\mu}{\sigma} \right) \right\}^{-1/\xi}, & \xi \neq 0 \\ e^{-\left( \frac{z-\mu}{\sigma} \right)}, & \xi = 0 \end{cases} \quad (1)$$

For $\xi \neq 0$, we have $t(z)^{1+\xi} = \left\{ 1 + \xi \left( \dfrac{z-\mu}{\sigma} \right) \right\}^{-(1+1/\xi)}$ and for $\xi = 0$, the $x$ domain restricted to

$\xi \left( \dfrac{z-\mu}{\sigma} \right) > -1$. The GEV distribution is parameterized using $\xi$, $\mu$, and $\sigma$ which are the shape,

location, and scale parameters, respectively and analogous to the skewness, mean and standard

deviation."

14- Line 272, this sentence may be improved.

Reply: Agreed. We have rewritten it as follows:

"The TWS difference between SAI and global warming in the region R2, particularly over the latter

part of the 21st century, is greater than for the rest of the domain."

15- Figure 3. The colors for legend may be improved. For example, use two different hues to represent

positive and negative values, and use white to repesent 0.

Reply: Implemented. As an example, please see new Figure 3.

[Figure]

**Figure 3.** Ensemble mean maps of TWS across the studied domain in the historical climate (a) over 1985-2014 and their projected future changes in the 2071–2100 period under the SSP5-85 GHG scenario (SSP5-8.5 minus historical (b) and GHG+SAI minus historical (c)). The extent to which the SAI impacts the TWS changes imposed by global warming is further shown (SAI minus SSP5-8.5 (d)). Hatched areas show where all ensemble members agree on the sign of the changes.

16- Figure 5, it may be much better to show empirical probabilities of observed TWS and visually show the performance of GEV distribution.

Reply: In response to your request, we have included graphs in the Supplementary Information for the three different scenarios in two regions, R2 and R5, as examples. Due to space constraints, it is not practical to display graphs for all three scenarios in all six regions.

[Figure]

**Figure RC2-8.** In region R2, the graphs illustrate the following scenarios: (a) historical, (b) global warming, and (c) the SAI scenario. In the left column, you can observe the relationship between empirical quantiles and model quantiles. In the right column, the graphs depict the probability density versus quantiles.

[Figure]

**Figure RC2-9.** As in Figure RC2-8 but for R5.

17- Figures 6 and 7. May add R-squared in the figures for better interpretation.

Reply: Implemented. Please see the new Figures 6 and 7 above.

---

## Author Response (AR1)

**To**: Ben Kravitz, Editor, ESD

**Subject**: Revision of Manuscript Reference Number EGUSPHERE-2023-1654

**Prof. Peter Haynes,**

Upon your recommendation, we have carefully revised the manuscript after addressing the comments and suggestions made by the reviewers. The following is the point–point response to all the comments (the comments are rewritten in black color and their corresponding replies in red). We appreciate the opportunity to revise our paper.

**Notice**: The line numbers refer to those in the marked version of the manuscript.

**Response to RC1:**

Rezaei et al. use climate model simulations to investigate the impacts of stratospheric aerosol injection (SAI) on the Middle East and North African (MENA) region. The study looks specifically at total water storage, and associated hydrology variables, in model simulations with SAI and climate change. MENA is an understudied region in the context of SAI, with important potential impacts on the water cycle, so this paper is a welcome addition to the literature. The paper overall needs reorganization and edits to the text for clarity, as well as modifications to the figures and explanation of statistical methods to better communicate the results. General comments are listed first, and further specific comments below.

Reply: We sincerely appreciate your effort and time in detailed reviewing our manuscript as well as constructive comments/suggestions. We have attempted to revise the manuscript in the light of your suggestions/comments. The following is also the point–point response to all the comments (comments are rewritten in black color and their corresponding replies in red).

**General comments:**

- I enjoyed reading about the study area in section 2.1 (in particular, the first two paragraphs). It would be informative to return to this context in the discussion section and talk about how the results in this paper might impact the regional climate more broadly.

Reply: Agreed. We have done this now in several places in the discussion (please see lines 472-474, 601, 608, and 631-632).

It would be helpful to give additional context for why these specific CESM simulations were used here (e.g., instead of the large ensemble simulations like GLENS/ARISE). I think a novelty of Tilmes et al. 2020 was the overshoot scenario but I don't believe those simulations were used here. I suggest adding some additional text in section 2.2 for context on the model simulations.

Reply: We have added additional text to section 2.2 (please see lines 232-260).

Generally, the paper does a good job specifying which comparison is being made (e.g., SSP relative to historical or SAI relative to historical) but there are some additional places to clarify the text in the results and discussion sections, which will make these sections easier to follow and interpret. I have some specific suggestions in the comments below.

The discussion section currently recaps/repeats many of the results (e.g., Lines 485-507). I suggest focusing on interpreting the results, highlighting particularly interesting results, and connecting with previous studies. Some of this is already present in the discussion section (e.g., Lines 438-442, 475-483, 509-518), so I think the section just needs some edits to move results into the results section and focus the discussion.

Reply: We have greatly revised the discussion. We moved some paragraphs into the results (please see lines 455-474). In addition, we revised some parts of the discussion (please see lines 601-608, 611-614, 628-643, 646-652, 662-664, 668-671, 674-675, and 709-712 in the tracked changes version).

**Specific comments:**

Lines 68-72: This paragraph seems out of place. Suggest moving this down to where projected future changes in the Mediterranean are discussed (e.g., Line 96).
Reply: Implemented.

Lines 74-84: Moving this paragraph down to after the discussion on climate change impacts (e.g., Line 108) would then transition to an introduction to SRM and associated impacts.
Reply: Implemented.

Line 90: Are there any more recent modeling studies (e.g., CMIP5 or CMIP6) that discuss projected changes in the MENA region? If not, it is worth pointing that out.
Reply: Here we focused on CMIP6 literature for consistency. However, we have added some new studies to the text (please see lines 93-99).

Line 102: An explanation of "soil moisture z-scores" is needed here.
Reply: Implemented (please see lines 115-116).

Line 162: This is covered in the introduction; suggest removing this sentence and moving the second sentence of this paragraph into the preceding paragraphs on the discussion of regional climate.
Reply: Implemented.

Line 174: In addition to defining "real evapotranspiration", please add an explanation on "potential evapotranspiration" and how that is calculated since that is also listed in Table 1. Is real ET a model output and potential ET is calculated from model output?
Reply: Real and potential ET have been calculated. We further added the definitions of potential and real EV to the text (please see lines 205-209).

Table 1: The caption mentions historical model output as the data source for this table – please add this to the text as well (e.g., Line 172). It might also make sense for Table 1 to come after the model is introduced in section 2.2.

Reply: We added it to the text as well (lines 203-204). However, we believe that Table 1 position in the Study area section is more appropriate.

Line 209: Please clarify the text here – what is meant by "in turn"?

Reply: It means "and consequently". We simply deleted this word.

Line 241: Was this correlation calculated or made by eye using the plots in Figure S2? And was the correlation tested for the other variable combinations? Calculating the correlations and reporting them in the paper would make for a stronger justification of the MLR model inputs.

Reply: We have calculated the correlation coefficients. Please see the following results. Except for region 5, the correlation coefficient between TWS and soil moisture in all regions under all scenarios and ensembles are larger than 0.90. The correlation coefficient between temperature and soil moisture in all cases is larger than 0.90 and statistically significant at the 5% level. However, we added these correlation values to the text (please see lines 328-338) and also included the following tables in the Supplementary Information.

**Table S2.** The average correlation between the variables under the available ensembles for global warming SSP5-8.5 scenario in the region R1. Consistently, the values inside the parenthesis are the difference-range values between minimum and maximum correlations. The insignificant correlation coefficients (p-value>0.05) are underlined.

| SSP-R1 | TWS | Temp | Precip | SM | RET | PET | LAI |
|--------|-----|------|--------|-----|-----|-----|-----|
| TWS | 1 | -0.35 (-0.31, -0.37) | 0.61 (0.58, 0.64) | 0.99 (0.98, 1) | 0.18 (0.15, 0.23) | -0.30 (-0.26, -0.32) | 0.77 (0.72, 0.78) |
| Temp | | 1 | -0.61 (-0.60, -0.62) | -0.35 (-0.32, -0.37) | 0.77 (0.76, 0.78) | 0.96 (0.96, 0.96) | 0.05 (0.02, 0.07) |
| Precip | | | 1 | 0.59 (0.56, 0.63) | -0.17 (-0.13, -0.19) | -0.56 (-0.55, -0.57) | 0.42 (0.40, 0.44) |
| SM | | | | 1 | 0.17 (0.14, 0.22) | -0.30 (-0.26, -0.32) | 0.74 (0.70, 0.77) |
| RET | | | | | 1 | 0.75 (0.75, 0.75) | 0.60 (0.58, 0.63) |
| PET | | | | | | 1 | 0.11 (0.08, 0.13) |
| LAI | | | | | | | 1 |

**Table S3.** As Table S2 but for the region R2.

| SSP-R2 | TWS | Temp | Precip | SM | RET | PET | LAI |
|--------|-----|------|--------|-----|-----|-----|-----|
| TWS | 1 | -0.25 (-0.24, -0.26) | 0.50 (0.47, 0.53) | 0.99 (0.99, 1) | 0.54 (0.50, 0.57) | -0.24 (-0.23, -0.25) | 0.72 (0.71, 0.74) |
| Temp | | 1 | -0.48 (-0.47, -0.50) | -0.24 (-0.22, -0.25) | 0.11 (0.10, 0.12) | 0.95 (0.94, 0.96) | -0.18 (-0.15, -0.20) |
| Precip | | | 1 | 0.49 (0.46, 0.51) | 0.53 (0.52, 0.55) | -0.51 (-50, -53) | 0.52 (0.51, 0.53) |
| SM | | | | 1 | 0.53 (0.49, 0.56) | -0.22 (-0.20, -0.24) | 0.71 (0.70, 0.73) |
| RET | | | | | 1 | 0.03 (0.02, 0.04) | 0.85 (0.84, 0.86) |
| PET | | | | | | 1 | 0.16 (0.14, 0.19) |
| LAI | | | | | | | 1 |

**Table S4.** As Table S2 but for the region R3.

| SSP-R3 | TWS | Temp | Precip | SM | RET | PET | LAI |
|---|---|---|---|---|---|---|---|
| TWS | 1 | -0.67 (-0.65, -0.70) | 0.51 (0.49, 0.52) | 0.98 (0.97, 0.99) | 0.34 (0.33, 0.35) | -0.54 (-0.52, -0.57) | 0.80 (0.78, 0.81) |
| Temp | | 1 | -0.83 (-0.82, -0.84) | -0.68 (-0.66, -0.70) | 0.23 (0.22, 0.24) | 0.97 (0.97, 0.97) | -0.44 (-0.42, -0.46) |
| Precip | | | 1 | 0.49 (0.46, 0.51) | -0.28 (-0.27, -0.30) | -0.82 (-0.81, -0.83) | 0.31 (0.29, 0.32) |
| SM | | | | 1 | 0.34 (0.33, 0.35) | -0.55 (-0.53, -0.57) | 0.80 (0.78, 0.81) |
| RET | | | | | 1 | 0.33 (0.32, 0.34) | 0.69 (0.68, 0.70) |
| PET | | | | | | 1 | -0.29 (-0.26, -0.31) |
| LAI | | | | | | | 1 |

**Table S5.** As Table S2 but for the region R4.

| SSP-R4 | TWS | Temp | Precip | SM | RET | PET | LAI |
|---|---|---|---|---|---|---|---|
| TWS | 1 | 0.08 (0.03, 0.11) | 0.19 (0.14, 0.23) | 0.99 (0.98, 1) | 0.35 (0.26, 0.42) | 0.08 (0.05, 0.10) | 0.53 (0.43, 0.59) |
| Temp | | 1 | -0.06 (-0.02, -0.11) | 0.08 (0.04, 0.11) | 0.04 (0.03, 0.06) | 0.92 (0.92, 0.92) | -0.07 (-0.02, -0.12) |
| Precip | | | 1 | 0.18 (0.13, 0.22) | 0.71 (0.68, 0.72) | -0.14 (-0.10, -0.20) | 0.25 (0.23, 0.27) |
| SM | | | | 1 | 0.32 (0.21, 0.40) | 0.08 (0.06, 0.10) | 0.53 (0.42, 0.59) |
| RET | | | | | 1 | -0.09 (-0.05, -0.13) | 0.62 (0.60, 0.64) |
| PET | | | | | | 1 | 0.04 (0.01, 0.06) |
| LAI | | | | | | | 1 |

**Table S6.** As Table S2 but for the region R5.

| SSP-R5 | TWS | Temp | Precip | SM | RET | PET | LAI |
|---|---|---|---|---|---|---|---|
| TWS | 1 | 0.15 (0.14, 0.17) | 0.19 (0.15, 0.23) | 0.74 (0.70, 0.76) | 0.29 (0.24, 0.33) | 0.19 (0.18, 0.21) | 0.33 (0.29, 0.35) |
| Temp | | 1 | 0.17 (0.12, 0.24) | 0.31 (0.24, 0.37) | 0.34 (0.31, 0.39) | 0.96 (0.96, 0.96) | 0.50 (0.46, 0.53) |
| Precip | | | 1 | 0.36 (0.31, 0.38) | 0.91 (0.88, 0.93) | 0.12 (0.07, 0.19) | 0.61 (0.54, 0.70) |
| SM | | | | 1 | 0.49 (0.41, 0.55) | 0.32 (0.25, 0.38) | 0.50 (0.43, 0.55) |
| RET | | | | | 1 | 0.30 (0.27, 0.36) | 0.82 (0.79, 0.85) |
| PET | | | | | | 1 | 0.47 (0.44, 0.50) |
| LAI | | | | | | | 1 |

**Table S7.** As Table S2 but for the region R6.

| SSP-R6 | TWS | Temp | Precip | SM | RET | PET | LAI |
|---|---|---|---|---|---|---|---|
| TWS | 1 | 0.29 (0.26, 0.31) | 0.29 (0.24, 0.35) | 0.98 (0.97, 1) | 0.46 (0.38, 0.53) | 0.26 (0.21, 0.31) | 0.26 (0.24, 0.31) |
| Temp | | 1 | 0.06 (0.02, 0.11) | 0.29 (0.26, 0.31) | 0.32 (0.27, 0.39) | 0.95 (0.94, 0.96) | 0.35 (0.28, 0.40) |
| Precip | | | 1 | 0.27 (0.22, 0.32) | 0.84 (0.82, 0.85) | 0.04 (0.02, 0.07) | -0.02 (0.00, -0.06) |
| SM | | | | 1 | 0.43 (0.36, 0.50) | 0.28 (0.26, 0.30) | 0.27 (0.24, 0.32) |
| RET | | | | | 1 | 0.26 | 0.05 |

| | | | | | (0.23, 0.30) | (0.03, 0.11) |
|---|---|---|---|---|---|---|
| PET | | | | | 1 | 0.22
(0.12, 0.29) |
| LAI | | | | | | 1 |

**Table S8.** As Table S2 but under SAI scenario for the region R1.

| SAI-R1 | TWS | Temp | Precip | SM | RET | PET | LAI |
|---|---|---|---|---|---|---|---|
| TWS | 1 | -0.20
(-0.18, -0.24) | 0.59
(0.58, 0.60) | 0.96
(0.95, 0.97) | 0.13
(0.13, 0.13) | -0.15
(-0.13, -0.18) | 0.52
(0.46, 0.59) |
| Temp | | 1 | -0.51
(-0.49, -0.52) | -0.06
(-0.02, -0.11) | 0.88
(0.87, 0.89) | 0.96
(0.96, 0.96) | 0.44
(0.38, 0.49) |
| Precip | | | 1 | 0.53
(0.52, 0.54) | -0.20
(-0.19, -0.21) | -0.47
(-0.46, -0.49) | 0.26
(0.23, 0.28) |
| SM | | | | 1 | 0.26
(0.25, 0.27) | -0.05
(0.00, -0.08) | 57
(50, 64) |
| RET | | | | | 1 | 0.89
(0.88, 0.90) | 0.76
(0.72, 0.78) |
| PET | | | | | | 1 | 0.52
(0.46, 0.57) |
| LAI | | | | | | | 1 |

**Table S9.** As Table S2 but under SAI scenario for the region R2.

| SAI-R2 | TWS | Temp | Precip | SM | RET | PET | LAI |
|---|---|---|---|---|---|---|---|
| TWS | 1 | -0.26
(-0.24, -0.28) | 0.53
(0.51, 0.55) | 0.97
(0.96, 1) | 0.54
(0.52, 0.57) | -0.25
(-0.23, -0.26) | 0.73
(0.69, 0.79) |
| Temp | | 1 | -0.53
(0.51, 0.55) | -0.24
(-0.22, -0.26) | 0.27
(0.26, 0.28) | 0.96
(0.96, 0.96) | -0.08
(-0.05, -0.09) |
| Precip | | | 1 | 0.51
(0.50, 0.53) | 0.41
(0.40, 0.42) | -0.57
(-0.54, -0.59) | 0.50
(0.48, 0.51) |
| SM | | | | 1 | 0.53
(0.51, 0.56) | -0.23
(-0.21, -0.24) | 0.72
(0.68, 0.78) |
| RET | | | | | 1 | 0.21
(0.21, 0.22) | 0.84
(0.84, 0.84) |
| PET | | | | | | 1 | -0.04
(-0.03, -0.05) |
| LAI | | | | | | | 1 |

**Table S10.** As Table S2 but under SAI scenario for the region R3.

| SAI-R3 | TWS | Temp | Precip | SM | RET | PET | LAI |
|---|---|---|---|---|---|---|---|
| TWS | 1 | -0.54
(-0.53, -0.55) | 0.43
(0.42, 0.44) | 0.98
(0.97, 0.99) | 0.25
(0.24, 0.26) | -0.46
(-0.46, -0.47) | 0.76
(0.75, 0.77) |
| Temp | | 1 | -0.78
(0.78, 0.79) | -0.54
(-0.53, -0.55) | 0.55
(0.54, 0.56) | 0.98
(0.96, 0.99) | -0.18
(-0.16, -0.19) |
| Precip | | | 1 | 0.42
(0.41, 0.43) | -0.37
(-0.35, -0.38) | -0.78
(-0.78, -0.78) | 0.18
(0.17, 0.21) |
| SM | | | | 1 | 0.26
(0.25, 0.26) | -0.46
(-0.46, -0.47) | 0.76
(0.75, 0.78) |
| RET | | | | | 1 | 0.59
(0.58, 0.59) | 0.67
(0.66, 0.67) |
| PET | | | | | | 1 | -0.09
(-0.07, -0.10) |
| LAI | | | | | | | 1 |

**Table S11.** As Table S2 but under SAI scenario for the region R4.

| SAI-R4 | TWS | Temp | Precip | SM | RET | PET | LAI |
|---|---|---|---|---|---|---|---|
| TWS | 1 | 0.03
(0.01, -0.04) | 0.28
(0.27, 0.30) | 0.99
(0.98, 1) | 0.45
(0.37, 0.52) | 0.03
(0.02, 0.04) | 0.56
(0.52, 0.60) |
| Temp | | 1 | -0.05
(-0.02, -0.07) | 0.03
(0.01, 0.04) | 0.09
(0.06, 0.10) | 0.94
(0.94, 0.94) | 0.16
(0.10, 0.24) |
| Precip | | | 1 | 0.27 | 0.78 | -0.14 | 0.32 |

| | TWS | Temp | Precip | SM | RET | PET | LAI |
|---|---|---|---|---|---|---|---|
| | | | | (0.26, 0.29) | (0.76, 0.79) | (-0.11, -0.15) | (0.26, 0.36) |
| SM | | | | 1 | 0.43 (0.36, 0.50) | 0.03 (0.02, 0.04) | 0.55 (0.51, 0.59) |
| RET | | | | | 1 | -0.02 (0.00, -0.03) | 0.68 (0.67, 0.69) |
| PET | | | | | | 1 | 0.15 (0.08, 0.23) |
| LAI | | | | | | | 1 |

**Table S12.** As Table S2 but under SAI scenario for the region R5.

| SAI-R5 | TWS | Temp | Precip | SM | RET | PET | LAI |
|---|---|---|---|---|---|---|---|
| TWS | 1 | 0.29 (0.22, 0.34) | 0.15 (0.14, 0.16) | 0.73 (0.64, 0.80) | 0.28 (0.26, 0.31) | 0.33 (0.27, 0.38) | 0.36 (0.28, 0.41) |
| Temp | | 1 | 0.12 (0.12, 0.13) | 0.29 (0.24, 0.34) | 0.36 (0.35, 0.37) | 0.97 (0.97, 0.97) | 0.56 (0.52, 0.59) |
| Precip | | | 1 | 0.37 (0.33, 0.41) | 0.88 (0.88, 0.89) | 0.09 (0.09, 0.10) | 0.56 (0.52, 0.58) |
| SM | | | | 1 | 0.54 (0.50, 0.58) | 0.32 (0.26, 0.37) | 0.50 (0.46, 0.54) |
| RET | | | | | 1 | 0.34 (0.32, 0.35) | 0.80 (0.77, 0.81) |
| PET | | | | | | 1 | 0.56 (0.52, 0.59) |
| LAI | | | | | | | 1 |

**Table S13.** As Table S2 but under SAI scenario for the region R6.

| SAI-R6 | TWS | Temp | Precip | SM | RET | PET | LAI |
|---|---|---|---|---|---|---|---|
| TWS | 1 | 0.11 (0.06, 0.17) | 0.26 (0.19, 0.31) | 0.99 (0.98, 1) | 0.44 (0.38, 0.50) | 0.11 (0.05, 0.16) | 0.33 (0.31, 0.35) |
| Temp | | 1 | 0.06 (0.05, 0.08) | 0.11 (0.06, 0.17) | 0.35 (0.33, 0.37) | 0.96 (0.96, 0.96) | 0.37 (0.34, 0.40) |
| Precip | | | 1 | 0.24 (0.18, 0.29) | 0.73 (0.72, 0.74) | 0.00 (0.00, 0.00) | 0.08 (0.03, 0.12) |
| SM | | | | 1 | 0.42 (0.36, 0.47) | 0.10 (0.05, 0.16) | 0.33 (0.31, 0.34) |
| RET | | | | | 1 | 0.29 (0.26, 0.31) | 0.26 (0.19, 0.30) |
| PET | | | | | | 1 | 0.32 (0.30, 0.35) |
| LAI | | | | | | | 1 |

Line 246: How often / how many outliers were removed using this method?
Reply: We totally fitted 36 models. The number of outliers removed is specific for each model. In some models, no outliers were detected while in others up to 5 data points were removed. We added the following sentence to the manuscript (lines 360-361):
"The number of outlier data points excluded varies from zero to 5 (over the 700 point) in the 36 models."

Line 254: Please add more details on the method here – specifically what is meant by "independent variable-order average over average contributions…" and "impacts adjusted for other regressors".
Reply: "independent variable-order average over average contributions…" refers to the fact that the order in which variables are added to the model can affect how much each variable contributes to the overall R-squared. If a variable that has a strong linear relationship with the

dependent variable be initially added, it may capture a significant portion of the variance, which can affect the subsequent contributions of other variables. To mitigate the order dependency, the LMG calculates average contributions by considering all possible orders and then averaging the results. This approach helps provide a more robust estimate of variable importance that is less sensitive to the specific order in which variables are considered. For clarity we revised this sentence as follows (please see lines 369-371):

"The LMG method considers the average contributions of each variable across different model sizes and then averages these averages to provide a more robust measure of variable importance."

To clarify the "impacts adjusted for other regressors" term we have added the following additional description to the text (please see lines 379-383).

Figure 2: It is difficult to see the trends in the anomalies with the strong seasonal cycle in certain regions. Suggest removing the seasonal cycle and/or some other method of filtering out noise here (e.g., running yearly means).

Reply: We have also extracted the long-term trends using Singular Spectrum Analysis (SSA) method (below figure), consistent with the results from Figure 2. This figure has also been included in the Supplementary Information (please see Fig. S5).

[Figure]

**Figure RC1-1.** The long-term trends of TWS anomaly relative to the TWS averaged over the historical period across MENA and the lands around the Caspian and Mediterranean Seas under global warming without (SSP5-8.5) and with SAI (SSP5-8.5-SAI). Figures a-f respectively are for regions R1 to R6. Shading in each curve shows the across-ensemble range. The dashed line crossing the *y*-axis at zero in each subplot is the ensemble mean of TWS over the historical period (1985-2014).

Lines 282-295: Please use the region labels (e.g., R1-R6) in the text here to ease the interpretation of Figure 3.

Reply: Implemented (please see new Figure 3).

Line 292: Clarify "Mean TWS" here – is that the temporal mean, ensemble mean, spatial mean, or some combination?
Reply: It refers to temporal-ensemble mean (please see lines 421 and 608).

Figure 3 (and others): For the difference plots (e.g., panels b-d), I recommend choosing a different color scale with a clear divergence at 0. With this yellow/blue color scale it is difficult to discern positive vs. negative regions of change. Same comment for Figures S2, S4. The color scale in Figure S1 works better for difference plots.
Reply: Implemented (please see new figures).

Lines 304-311: Use percentage values instead of absolute kg/m^2 changes in the text here to match the black labels in Figure 4. Or use absolute labels in Figure 4, which would match the y-axis.
Reply: We have used percentage in the text in the new version (please see lines 435-436).

Figure 4: In addition to the partial reversals (R1, R3, R4) and the overcompensation (R2), SAI also has an amplifying effect in R5 and a slight overcompensation in R6 – it is worth noting these responses in the text (even if to say they are not significant).
Reply: The following sentence has been added to the text:
"SAI also has an amplifying effect in R5 and a slight overcompensation in R6, but its impact is statistically insignificant."

Figure 4: Why are there three p-values shown at the bottom of each panel? I assume two of the values denote the significance of the changes in SSP and SAI relative to historical, but what does the other value represent? Please clarify in the figure caption. Same comment for Figures S3 and S5.
Reply: Implemented (please see new captions for Figs. 4, S7, and S8).

Line 330: Similar question to Line 241 – was significance calculated here or by eye? Why do non-overlapping curves imply significance?
Reply: We first conducted the repeated measures analysis of variance which compares means across one or more variables that are based on repeated observations, and then performed post hoc Tukey-Kramer comparisons to determine which curves (including its upper and lower bounds) are significantly different from each other (please see the new Figure 5 and its caption). However, we have added the above explanations to the text (please see 492-495).

Lines 344-360: Please use the region labels R1-R6 in text here to ease comparison to Table 2.
Reply: Implemented (please see lines 513-522). Please see the revised version of this part copied below:

Line 346 and following: Please clarify "decreases the TWS extremes" – does this mean a decrease in positive extremes (i.e., fewer wetter conditions) or negative extremes (fewer drier conditions) or both?

Reply: Note that return levels refer to the peaks in the TWS time series. Since here we used the anomalies relative to the historical mean, the peaks may be negative or positive. However, here we mean the fewer wetter conditions. Please see the following revised part (lines 511-512): "Global warming, on the whole, decreases the TWS extremes (i.e., fewer wetter conditions) at 30- to 100-year return periods …".

Lines 378-380: Please clarify here whether this is referring to the most important variable under SAI or SSP (or both).
Reply: Under both SSP and SAI.

Lines 386-389: Please clarify what is meant by "due to evapotranspiration" if this is looking only at temperature and precipitation ("with just temperature and precipitation as independent variables"). Are there results that look at subsets of these three variables and are they included somewhere?
Reply: Our mean of "With just temperature and precipitation as independent variables, …" is that if we compare the temperature importance role on TWS with precipitation role. We just want to know between precipitation and temperature which one is more important. However, in all TWS MLR models, all four variables of real ET, precipitation, temperature, and leaf area index (i.e., vegetation coverage) have been considered.

Line 399: Please include the specific variance explained values for the MLR models somewhere in the text or figures (e.g., the bars of Figure 6-7).
Reply: We have included the ensemble-mean variance on Figures 6-7.

Figures 6-7: Here, or perhaps in the methods section, please provide some context for the importance values (y-axis). Is this unitless, and if so, should the individual variable contributions total to 1 if all the appropriate variables were sampled? Are interactions considered?
Reply: Implemented (please see new Figures 6 and 7 above). Importance is a unitless variable and the sum of all independent variable importance's in each model equals the model's variance explained. The importance values for individual variables do not necessarily need to total to 1. However, we have added the following sentence to the text (lines 383-385):
"Importance is a unitless variable and the sum of all independent variable importance's in each model equals the model's explained variance."
In the case of LMG, it's a method that explicitly considers interaction effects and decomposes the total variance explained into contributions from individual variables, pairs of variables, and higher-order interactions. Therefore, LMG importance values can provide insights into both main effects and interaction effects.

Lines 444-457: Most of this paragraph should go in the results section, as the supplemental figures have not yet been discussed. The last sentence gets to a comparison with other studies which is appropriate for the discussion section and can be merged with another paragraph.
Reply: We have moved this part into the results, last part of section 3.1.

Line 446: Please specify which simulation "The TWS decreasing patterns" refers to.
Reply: Both SSP5-8.5 and SSP5-8.5-SAI scenarios.

Line 461: Related to vegetation, it is worth discussing the competing impacts of high CO2 and less solar radiation in the SAI scenario. These impacts could also be contributing to the overall ET, soil moisture, and TWS responses. The regions discussed here have varying amounts of vegetation and that could be contributing to the range of regional responses.

Reply: It has been found that considering vegetation variable for TWS leads to improved TWS model output (Trautmann et al., 2022). Plants absorb water from the soil and release it into the atmosphere through transpiration. Hence, we added leaf area index (LAI) as a new variable into MLR models and results (please see new Figures 6 and 7 above and Figures S6 and S7 below). We also used LAI findings in the results and discussion sections (please see lines 324, 549-552, 571-576, 638-639, and 646-652).

[Figure]

**Figure S6**. As Figure 3 but for LAI.

[Figure]

**Figure S7**. As Fig. 4 but for LAI.

Figure S3: I thought the middle row of this plot (TWS) would be same as Figure 4, but it appears to be different. What is plotted here and what is the difference with Figure 4?
Reply: Agreed. We rechecked and replotted Figures S3 and 4, now they are the same. It seems that in the previous version of Figure S3 was an initial plot after which I edited some parts of the code.

Figure S4: For the middle row (temperature) difference plots, the color bar limits should be increased on both ends to better show the regional responses.
Reply: Various limits have been tested; the following is the best.

[Figure]

**Figure S4**. As in Figure S2, but for the variables of precipitation (upper row), surface temperature (middle row), and real evapotranspiration (ET, bottom row).

Data availability: Suggest providing some more information on how to access these specific CESM simulations via the ESGF website (e.g., Source ID, Experiment ID). Tilmes et al. 2020 also has a DOI for the SAI simulations which should be included if those experiments are not on ESGF: https://doi.org/10.26024/t49k-1016.
Reply: we rewrite the data availability section accordingly.

Technical corrections:
Line 39: Typo "Projected" should not be capitalized.
Reply: Implemented.

Lines 335-336: I think this should be "return levels" instead of "level returns".
Reply: Implemented.

Lines 335-337: Should these sentences be combined?
Reply: Implemented.

Figure 5: Please add panel labels to the subplots and update caption to "(a to f)".
Reply: Done (please see new Figure 5).

Line 397: Typo "EV"
Reply: Corrected.

Line 467: Typo "EV"
Reply: Corrected.
Lines 520-522: I think "SAI" is missing after "with…and without" here.
Reply: Added.

**Response to RC2:**

Thank you for inviting me for reading this article. The authors evaluate terrestrial water storage under SSP5-8.5 and SSP5-8.5-SAI scenarios across the Middle East and North Africa. The results are useful for supporting aerosol intervention strategy against global warming and water resources management for Mediterranean, Middle East, and North Africa. I have some concerns about the methods and figures which may be helpful for improvement.

Reply: We sincerely appreciate your effort and time in reviewing our manuscript as well as your constructive comments/suggestions. We have made every effort to incorporate your feedback effectively. Below, you will find a detailed response to each comment, with comments presented in black and our responses in red.

1- Section 2.3. The authors calculate return periods from GEV distribution. However, GEV distribution is used to simulate maximum value in a certain period, instead of monthly values. The authors may give more details about how to apply GEV distribution. Did the authors calculate the annual maximum TWS values? In addition, authors may provide empirical probabilities and examine whether annual maximum TWS follows GEV distribution or other distributions.

Reply: We applied a GEV distribution to the complete dataset of monthly TWS values without explicitly setting maximum values. This approach allowed us to estimate the parameters of the GEV distribution using the entire dataset. However, in response to your request, we have also extracted the annual maximum TWS values and provided the corresponding fitted GEV distribution for comparison with the full dataset scenario (e.g., Figs. S11 and S12).

Overall, the probability densities for both datasets exhibit a high degree of similarity across various regions and scenarios. For instance, Figures S11 and S12 illustrate the probability densities for the R2 and R5 regions. Additionally, the graphs depicting return levels versus return periods based on annual maximums (Fig. S13) closely resemble the results obtained from the entire dataset (Fig. 5). In all cases, the trends are highly similar (compare Figure 5 and Figure S13), although it's worth noting that the annual maximums scenario exhibits slightly wider upper and lower bounds compared to the entire dataset scenario. Regarding the significance test for differences between historical, global warming, and SAI scenarios, the results are consistent across all cases. However, there is an exception in the case of the difference between historical and SAI scenarios in R5. In the entire dataset scenario, a significant difference is observed (Fig. 5e), whereas in the annual maximums case, it does not reach significance (Fig. S13e).

In light of these explanations, we have retained the results obtained from the entire dataset in the main text, and we included the results from the annual maximums scenario in the Supplementary Materials.

[Figure]

**Figure S11.** Probability density curves for Region R2, comparing two scenarios: one using all available data (left column) and the other using annual maximum values (right column) under the historical conditions (upper row) as well as the GHG emissions (middle row) and SAI (lower row) scenarios for region R2.

[Figure]

**Figure S12.** As Figure S11 but for the region R5.

[Figure]

**Figure S13.** As in Figure 5 but for the annual maximums.

2- Line 211. The historical period is from 1985-2014, and future period is from 2071-2100. The authors do not analyze mid-21th century. The authors may explain why you do not analyze the full period from 1850-2100.

Reply: Agreed (please see lines 283-294).

3- Authors only select CESM2 for analysis. The authors may evaluate the performance CESM2 for historical climate over the study area to validate this model.

Reply: We have incorporated new information into the new version (please see lines 245-260).

4- Authors use the MLR model to predict TWS. Apart from potential ET, the actual ET is also correlated with temperature and precipitation. How to solve the collinearity between ET, temperature and precipitation?

Reply: In assessing collinearity, we employed the VARCLUS procedure, a method that partitions a set of numeric variables into distinct or hierarchical clusters (Sarle, 1990). Each cluster is associated with a linear combination of the variables it contains. The criterion in this procedure is that if the proportion of the variance explained by a cluster is larger than 0.8 (Figs. S3 and S4), we should choose one variable from that cluster. It's worth noting that there was minimal variation among ensemble members for each scenario across regions. As a result, we have exclusively presented results for the ensemble r1 in Figs. S3 and S4. Based on our findings (refer to Figs. S3 and S4), in most instances, we needed to select one variable from pairs like potential ET and temperature or TWS and soil moisture. Consequently, we opted for temperature and TWS for our analysis. However, in the arid regions R4 to R6, although both real ET and precipitation were categorized within a single cluster, the proportion of variance explained by the cluster fell below 0.8. Hence, we decided to consider both variables. In response to the comment made in RC1, we also included leaf area index (LAI) as an additional variable in our analysis.

We have Incorporated above additional explanations into Section 2.4 of the methodology.

[Figure]

**Figure S3.** This tree diagrams illustrate the cluster hierarchy within ensemble r1 of the SSP5-8.5 scenario across regions R1 to R6. The y-axis represents the Proportion of Variance Explained.

[Figure]

**Figure S4.** As in Figure S3 but for the SSP5-8.5-SAI scenario.

5- Authors remove outliers in the MLR model. This will artificially give better results. Please justify the removal of these values?

Reply: Overall, the maximum number of outliers removed (5) is relatively insignificant when considering the total number of records in each timeseries (which exceeds 700 in our study). Therefore, it is unlikely to have a substantial impact on the model. Nonetheless, we have incorporated the following statement into the text (lines 360-361):

"The number of outlier data points excluded varies from zero to 5 (over the 700 point) in the 36 models."

6- The temporal autocorrelation is an important component in TWS evolution. Monthly TWS is not only impacted by concomitant precipitation and temperature, but also antecedent soil moisture and climatic variables. Authors may consider include climatic variable in previous months as predictors as well.

Reply: In our models, we excluded soil moisture from the list of predictor variables due to its collinearity with TWS. Additionally, we conducted a temporal autocorrelation analysis (As an example, Figs. RC2-1 and RC2-2 below) on all the variables, including temperature, precipitation, real ET, and LAI data for each model. This analysis was carried out using the Autocorrelation function at a 95% confidence level.

In all regions (except R4), the autocorrelation results indicated that the lags at the first and second months were statistically significant, while the third month lag was almost non-significant. Therefore, we modified the LMS model to include information from the two preceding months in these regions.

However, in region R4, we observed different patterns. In this region, both real ET and temperature significantly depended on their respective conditions from the two previous months, while precipitation did not show this effect. Moreover, TLAI in R4 exhibited dependencies on the first three and four preceding months under the SSP and SAI scenarios, respectively. Consequently, we incorporated specific lagged months for each variable in R4.

We have included the updated figures (Figures 6 and 7) to reflect these changes. Furthermore, we revised the MLR methodology and Section 3.3 in accordance with this information (please see lines 328-351).

[Figure]

**Figure RC2-1**. The autocorrelation plot for real ET in region R4 under the SAI scenario, specifically ensemble member 003. The y-axis represents lag values in terms of months.

[Figure]

**Figure RC2-2.** As in Figure RC2-1 but for TLAI.

[Figure]

**Figure 6.** LMG importance plot (Lindeman et al., 1980) of the four independent variables in the regression for TWS for the global warming SSP5-8.5 scenario in each region. The bar and range-bar respectively show the ensemble mean importance and the range of importance from the three ensemble members. The three values in red on each subplot shows the minimum, mean, and maximum variances explained by models.

[Figure]

**Figure 7.** As in Fig. 6, but for the SSP5-8.5-SAI scenario.

7- Water storage include soil moisture, groundwater, snow, ice, and others. Figure S3 seems to indicate soil moisture is the dominant driver of TWS variations. It may be insightful for evaluate the relative contributions of other components in TWS.

Reply: In the CMIP6 climate models, TWS is defined as the sum of snow water equivalent and soil moisture (Wu et al., 2021). Consequently, it is reasonable to assume that soil moisture plays a dominant role in driving TWS variations, especially in arid regions. We have incorporated this information into the text (please see lines 272-273 and 460-461).

8- Figure S4 is important for interpreting current results. May consider to place this figure in main text.

Reply: Understood. Since the primary focus of the study is on Terrestrial Water Storage (TWS), it will be kept in the supplementary materials.

9- It may be useful to compare the results with previous evaluations (https://www.nature.com/articles/s41558-020-00972-w; Global terrestrial water storage and drought severity under climate change).

Reply: Implemented. We used it in our discussions (please see lines 122-124, 291-293, 460-461, and 604-606).

10- Line 29, this sentence may be improved. May explain "more continental" and "hyper-arid" climates? Specify what is different response?

Reply: Implemented (please see lines 212-215). To clarify the distinctive response observed in R5, we included a new sentence into the abstract (please see lines 30-32).

11- Line 86, may place this paragraph earlier than the introduction of SRM, which is proposed to address climate change.

Reply: Implemented.

12- Line 127. What is the regional consequence and hydrological cycle? May give more explanations

Reply: We revised it (please see lines 157-158).

13- Line 228 and Eq. (1). The authors give the equations for Xi = 0 in equation (2). It may be better to provide CDF when Xi ($\xi$)= 0 in Equation (1) as well. In addition, I think Eq.(1) is the CDF instead of PDF. It is better to clearly specify this.

Reply: Implemented (please see lines 304-307).

14- Line 272, this sentence may be improved.

Reply: Agreed. We have rewritten it as follows:

"The TWS difference between SAI and global warming in the region R2, particularly over the latter part of the 21st century, is greater than for the rest of the domain."

15- Figure 3. The colors for legend may be improved. For example, use two different hues to represent positive and negative values, and use white to repesent 0.

Reply: Implemented.

16- Figure 5, it may be much better to show empirical probabilities of observed TWS and visually show the performance of GEV distribution.

Reply: In response to your request, we have included graphs in the Supplementary Information for the three different scenarios in two regions, R2 and R5, as examples (Figs. S1 and S2). Due to space constraints, it is not practical to display graphs for all three scenarios in all six regions.

[Figure]

**Figure S1.** In region R2, the graphs illustrate the following scenarios: (a) historical, (b) global warming, and (c) the SAI scenario. In the left column, you can observe the relationship between empirical quantiles and model quantiles. In the right column, the graphs depict the probability density versus quantiles.

[Figure]

**Figure S2.** As in Figure S1 but for R5.

17- Figures 6 and 7. May add R-squared in the figures for better interpretation.

Reply: Implemented (Please see the new Figures 6 and 7).

---

## Author Response (AR2)

**Prof. Ben Kravitz,**

Thank you immensely for accepting our paper (EGUSPHERE-2023-1654) for publication in ESD as a highlight paper. We have made the requested changes, including adding the corresponding author's name and email address below the affiliations. Additionally, we have formatted the reference list to align with the standard format of your journal.